# SuperVLAD: Compact and Robust Image Descriptors for Visual Place Recognition

**Feng Lu**[1,2*]   **Xinyao Zhang**[1*]   **Canming Ye**[1*]   **Shuting Dong**[1,2]
**Lijun Zhang**[3]   **Xiangyuan Lan**[2,4†]   **Chun Yuan**[1†]

[1]Tsinghua Shenzhen International Graduate School, Tsinghua University
[2]Pengcheng Laboratory   [3]CIGIT, Chinese Academy of Sciences
[4]Pazhou Laboratory (Huangpu)
{lf22@mails,yuanc@sz}.tsinghua.edu.cn   lanxy@pcl.ac.cn

## Abstract

Visual place recognition (VPR) is an essential task for multiple applications such as augmented reality and robot localization. Over the past decade, mainstream methods in the VPR area have been to use feature representation based on global aggregation, as exemplified by NetVLAD. These features are suitable for large-scale VPR and robust against viewpoint changes. However, the VLAD-based aggregation methods usually learn a large number of (*e.g.*, 64) clusters and their corresponding cluster centers, which directly leads to a high dimension of the yielded global features. More importantly, when there is a domain gap between the data in training and inference, the cluster centers determined on the training set are usually improper for inference, resulting in a performance drop. To this end, we first attempt to improve NetVLAD by removing the cluster center and setting only a small number of (*e.g.*, only 4) clusters. The proposed method not only simplifies NetVLAD but also enhances the generalizability across different domains. We name this method **SuperVLAD**. In addition, by introducing ghost clusters that will not be retained in the final output, we further propose a very low-dimensional **1-Cluster VLAD** descriptor, which has the same dimension as the output of GeM pooling but performs notably better. Experimental results suggest that, when paired with a transformer-based backbone, our SuperVLAD shows better domain generalization performance than NetVLAD with significantly fewer parameters. The proposed method also surpasses state-of-the-art methods with lower feature dimensions on several benchmark datasets. The code is available at https://github.com/lu-feng/SuperVLAD.

## 1   Introduction

Visual Place Recognition (VPR) is a task that aims at quickly estimating the coarse geographical location of a place image (*i.e.*, query) by retrieving the most similar images from a geo-tagged database [10, 41]. It has garnered significant attention in both computer vision and robotics communities, driven by its wide applications in augmented reality [42] and robot localization [59], etc. However, VPR still faces some challenges: On the one hand, images captured at the same place may vary dramatically due to the changes in conditions (*e.g.*, lighting and weather) and viewpoints. On the other hand, multiple different places can show high similarity, which may lead to perceptual aliasing [36]. It is quite challenging to address these issues simultaneously, especially for VPR methods that use compact global descriptors to represent the place images.

---

*Equal contribution.

†Corresponding authors.

38th Conference on Neural Information Processing Systems (NeurIPS 2024).

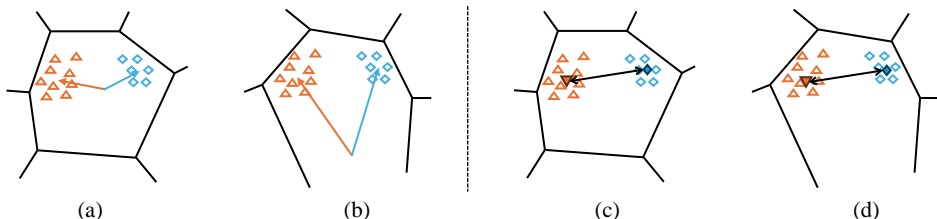

Figure 1: VLAD and SuperVLAD similarity measures under different clusterings (Voronoi cells). Orange triangles and blue diamonds depict local descriptors from two different images. In (a) and (b), orange and blue arrows are the sum of residuals (for VLAD). With different training data distributions, the different cluster centers are yielded, causing opposite similarity results using cosine similarity (or normalized L2 distance). Compared to VLAD, our SuperVLAD, as shown in (c) and (d), simply calculates the distance between the weighted sum of local features directly, freeing from the impact of cluster centers. Thus, only minor changes will occur when dealing with two different distributions.

The VPR is typically implemented with the image retrieval approaches [4, 48]. The place images are first described with global features and then the nearest neighbor search is performed over the database to get the matched place images of the query. The global descriptors are usually obtained by applying the aggregation methods, such as Bag of Words [3] and Vector of Locally Aggregated Descriptors (VLAD) [28, 35, 31, 53], to process local (or patch) features. Although some hierarchical (*i.e.*two-stage) VPR methods improve robustness under challenging environments by cross-matching local features in the query and candidate images for re-ranking, they incur considerable additional computational latency and memory footprint. Taking a step back, the first stage of these hierarchical methods still requires the use of global features to retrieve candidates. As such, representing place images with compact and robust global descriptors is always the most essential and important issue in VPR.

In the VPR methods based on neural networks, the NetVLAD [4] and GeM pooling [48] are the most commonly used aggregation/pooling methods. The former aggregates the feature maps extracted by neural networks with a trainable generalized VLAD layer. Its main difference from the vanilla VLAD [28] lies in its differentiable soft assignment and better performance. However, it still constructs the output vector by summing the residuals of descriptors assigned to each cluster (like VLAD), which is calculated by subtracting the learned cluster center from the local descriptors. When training data distributions differ, the learned cluster centers also vary. Even if we use a well-generalized backbone to extract the local features of two images, the differences in cluster centers can lead to completely different results in the similarity of the output global features (see Fig. 1). In other words, when there is a domain gap between the training and inference data, the cluster center learned on the training set is usually not suitable for inference, which can cause performance drops [5]. Besides, the global descriptor yielded by NetVLAD has a large dimension. Even with dimensionality reduction methods like PCA (or other learnable linear projections), the dimension often remains notably higher compared to the output of pooling methods. As a result, some VPR approaches requiring very low-dimensional features use GeM pooling to yield global descriptors at the cost of certain performance.

To address the above issues, we propose a compact and robust global descriptor for VPR, named SuperVLAD. Our method uses the transformer-based backbone to produce initial patch features. Similar to NetVLAD, SuperVLAD calculates the weights of assigning patch features to clusters in the soft-assignment way. The key difference is that SuperVLAD does not compute cluster centers. Instead, we directly perform a weighted summation of the local features assigned to each cluster to achieve the purpose of aggregating local features. This means SuperVLAD only needs to learn the assignment of local features to clusters, without needing to learn the cluster centers, which directly improves its domain generalization. Meanwhile, we set a small number of clusters (an order of magnitude lower than vanilla NetVLAD), resulting in more compact global descriptors. Furthermore, by introducing supernumerary ghost clusters during the soft assignment and retaining one real (useful) cluster in the final output vector, we obtain a 1-cluster VLAD descriptor with extremely low dimensionality and promising performance. Our work brings the following **contributions**:

(1) We propose a SuperVLAD aggregation method that does not require cluster centers to produce robust global descriptors. It can mitigate the performance degradation in NetVLAD caused by the training and testing data bias, while having fewer parameters (*i.e.*, more lightweight). Besides, it can use only a small number of clusters to yield compact descriptors.

(2) We also design a 1-cluster VLAD method by employing supernumerary ghost clusters during the soft assignment. It produces very low-dimensional features similar to the pooling methods. Compared to the same-dimensional GeM feature or class token, it shows notable performance advantages.

(3) We conduct extensive experiments using various transformer-based backbones, demonstrating the effectiveness of our method. Furthermore, our SuperVLAD with the DINOv2 backbone outperforms state-of-the-art (SOTA) methods on several benchmark datasets with lower-dimensional descriptors.

## 2   Related Work

In the early development of VPR, the methods predominantly relied on hand-crafted local features such as SURF [6, 13]. These local descriptors were then aggregated into global features using algorithms such as Bag of Words [3], Fisher Vector [47], and VLAD [28, 53, 5, 32]. The resulting global feature vectors were subsequently used to perform a nearest neighbor search over the database to retrieve the most similar images. As deep learning techniques have advanced, the representation capability of deep features has been widely recognized in the VPR community [51, 4, 29, 11, 12, 44, 18, 20, 58, 60, 52, 33, 1, 2, 21, 7, 10]. Arandjelovic et al.[4] designed a pioneering architecture that combines deep neural networks with the proposed differentiable VLAD aggregation approach called NetVLAD. Likewise, other traditional aggregation algorithms were also transformed into differentiable modules as the aggregation layer of neural networks for end-to-end training [26, 45, 61]. Despite remarkable performance over global max (or average) pooling, NetVLAD-related methods [4, 29, 34, 21] tended to have high-dimensional feature representations, which can be a drawback in terms of real-time performance [9]. To address this, the Generalized Mean (GeM) pooling [48] was considered a simple alternative that obtains low-dimensional global features. This method simply extended global average pooling by using the $p$-norm of local features instead of the average, where $p$ is also a trainable parameter. Additionally, Berton et al.[9] established an open-source benchmark to fairly compare these standard global-retrieval-based VPR approaches under a unified framework.

Although global-retrieval-based methods have achieved reasonable performance, most of them exhibit limited robustness in challenging environments and are susceptible to perceptual aliasing. Two common methods to improve the robustness of VPR systems are leveraging temporal consistency constraints and spatial consistency constraints. The first method matches image sequences (*i.e.*, maintains temporal continuity) [43, 16, 22, 37, 19] to achieve robust VPR with extreme condition changes. The second method usually involves a two-stage VPR process [25, 24, 8, 55, 40, 67, 38, 41], where it first retrieves the top-k candidate images in the database with global features and then re-ranks these candidates by matching local features with spatial consistency verification. However, these methods always introduce extra consumption in computational latency or/and memory footprint.

In the last two years, with the emergence of purpose-built large-scale VPR training datasets [7, 1] and pre-trained foundation models [62, 46], some robust global-retrieval-based VPR methods have been proposed. CosPlace [7] and EigenPlaces [10] cast the training of VPR as a classification problem and trained the VPR model on the San Francisco eXtra Large (SF-XL) datasets. MixVPR [2] incorporated the deep features with the multi-layer perceptrons and trained the model with the Multi-Similarity loss [56] on the large-scale and appropriate supervised GSV-Cities [1] dataset. These works all have achieved outstanding performance using only the CNN models. Other works [30, 41, 39, 27] were based on visual foundation models and achieved better results. DINOv2 [46] has been the most widely used foundation model in VPR. It is a ViT-based [17] model trained on the large-scale curated LVD-142M dataset using a self-supervised strategy and can offer powerful visual features for downstream tasks. SelaVPR [41] proposed a hybrid global-local adaptation method to adapt the DINOv2 model for two-stage VPR. CricaVPR [39] also adapted DINOv2 as the backbone and proposed a cross-image correlation-aware representation learning method to enhance the robustness of image features. A closely related work to our SuperVLAD is the SALAD work [27], which used DINOv2 as the backbone and presented a novel aggregation algorithm to improve NetVLAD. SALAD redefined the soft assignment of local features in NetVLAD as an optimal transport problem and employed the Sinkhorn algorithm [14] to solve it. Different from this work, our SuperVLAD maintains the soft-assignment way of NetVLAD but effectively solves the issue caused by the learned cluster centers in NetVLAD being unsuitable for various inference data distributions.

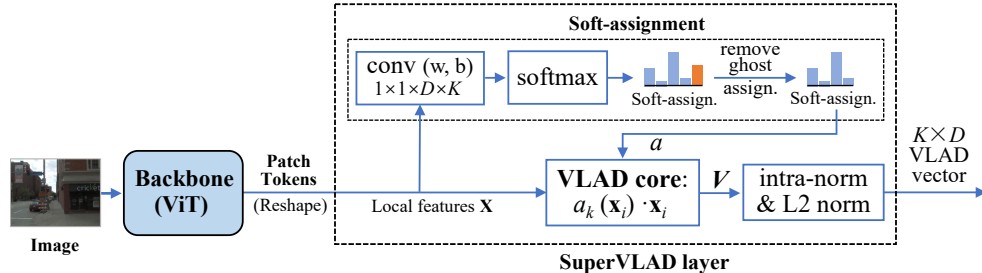

Figure 2: Illustration of the proposed SuperVLAD layer. It aggregates the patch tokens output by the transformer-based backbone and produces a $K \times D$ vector as the global descriptor. Note that the VLAD core of SuperVLAD has no cluster center, which is the main difference from NetVLAD.

## 3 Methodology

Fig. 2 shows the overview of SuperVLAD. We first use a backbone to extract local features (Sec. 3.1), then aggregate them with SuperVLAD (Sec. 3.2). 1-Cluster VLAD is an alternative to get more compact descriptors (Sec. 3.3) and cross-image encoder is optional to boost performance (Sec. 3.4).

### 3.1 Local Features Extraction

The Vision Transformer (ViT) [17] and its variants possess superiority in capturing long-range feature dependencies and have shown remarkable performance on various computer vision tasks [66, 63, 64], including VPR [55, 41]. In this work, we use ViT (or its variants) as the backbone for feature extraction. ViT initially divides the image into $N$ patches and linearly projects them into $D$-dim patch embeddings $x_p \in \mathcal{R}^{N \times D}$, after which it prepends a learnable class token to $x_p$ as $x_0 \in \mathcal{R}^{(N+1) \times D}$. Following the addition of positional embeddings, $x_0$ is input into a sequence of transformer encoder layers to generate the feature representation. The final output of ViT includes one class token and $N$ patch tokens. We directly discard the former and use the latter as local features to be input to the subsequent SuperVLAD aggregation layer, thereby obtaining the final global descriptor of the place image.

### 3.2 SuperVLAD Layer

SuperVLAD produces the global descriptor of an image by first associating all local features of this image with $K$ clusters and subsequently aggregating these features into each cluster. The basic process of the SuperVLAD layer is similar to that of NetVLAD [4], so we start by introducing our work from NetVLAD (and VLAD [28]).

For NetVLAD, given $N$ local descriptors $\{\mathbf{x}_i\}$ ($\mathbf{x}_i \in \mathcal{R}^D$) of an image as input, and $K$ cluster centers $\{\mathbf{c}_k\}$ ($\mathbf{c}_k \in \mathcal{R}^D$) as parameters, it computes a matrix $V \in \mathcal{R}^{K \times D}$ as the image representation. The $k$-th row in the matrix $V$ accumulates the (weighted) residuals $(\mathbf{x}_i - \mathbf{c}_k)$ of local descriptors assigned to cluster $\mathbf{c}_k$. More formally, the $(k, j)$-th element in the matrix $V$ is computed as follows:

$$V_{k,j} = \sum_{i=1}^{N} a_k(\mathbf{x}_i)(x_{i,j} - c_{k,j}),$$ (1)

where $x_{i,j}$ and $c_{k,j}$ are the $j$-th element of the $i$-th local descriptor and $k$-th cluster center, respectively. $a_k(\mathbf{x}_i)$ is the weight of the local descriptor $\mathbf{x}_i$ assigned to the cluster $\mathbf{c}_k$. In VLAD, the assignment is hard, i.e., $a_k(\mathbf{x}_i)$ equals 0 or 1. In contrast, NetVLAD replaces it with the soft assignment and computes $a_k(\mathbf{x}_i)$ as:

$$a_k(\mathbf{x}_i) = \frac{e^{-\alpha\|\mathbf{x}_i - \mathbf{c}_k\|^2}}{\sum_{k'} e^{-\alpha\|\mathbf{x}_i - \mathbf{c}_{k'}\|^2}},$$ (2)

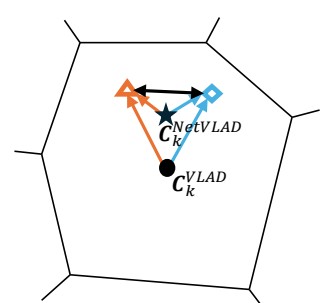

Figure 3: Unlike VLAD, since the parameters $\mathbf{w}_k$ and $b_k$ used for soft-assignment in NetVLAD are decoupled from cluster center $\mathbf{c}_k$, $\mathbf{c}_k$ does not necessarily coincide with the true centroid of the cluster (Voronoi cell). Its robustness against domain shift can be improved to some extent. SuperVLAD completely eliminates the need for cluster centers and avoids their negative impact.

Expanding the squares in Eq. 2, we can readily observe the cancellation of the term $e^{-\alpha\|\mathbf{x}_i\|^2}$ between the numerator and the denominator, yielding the following form:

$$a_k\left(\mathbf{x}_i\right) = \frac{e^{\mathbf{w}_k^T \mathbf{x}_i + b_k}}{\sum_{k'} e^{\mathbf{w}_{k'}^T \mathbf{x}_i + b_{k'}}}, \tag{3}$$

where vector $\mathbf{w}_k = 2\alpha\mathbf{c}_k$ and scalar $b_k = -\alpha\|\mathbf{c}_k\|^2$. In the implementation of NetVLAD, $\{\mathbf{w}_k\}$, $\{b_k\}$, and $\{\mathbf{c}_k\}$ are set as three independent sets of trainable parameters (while VLAD has only $\{\mathbf{c}_k\}$). That is, $\{\mathbf{w}_k\}$ and $\{b_k\}$ are actually decoupled from $\{\mathbf{c}_k\}$. As shown in Fig. 3, NetVLAD, with its greater flexibility than VLAD, has some potential to alleviate the performance drop caused by domain shift [4]. However, NetVLAD retains cluster centers, which would be hardly suitable for various data distributions in inference as it is learned on one training set [5]. Since the assignment of local features has been decoupled from the cluster centers in NetVLAD (in VLAD, it is based on the distance to the cluster center), we can directly remove the cluster center and aggregate the first-order statistics of local features (rather than residuals) assigned into each cluster. So, the matrix $V$ in SuperVLAD can be formulated as

$$V_{k,j} = \sum_{i=1}^{N} a_k\left(\mathbf{x}_i\right) x_{i,j} = \sum_{i=1}^{N} \frac{e^{\mathbf{w}_k^T \mathbf{x}_i + b_k}}{\sum_{k'} e^{\mathbf{w}_{k'}^T \mathbf{x}_i + b_{k'}}} x_{i,j}. \tag{4}$$

For each cluster, SuperVLAD has the parameters $\mathbf{w}_k$ and $b_k$[1], compared to NetVLAD with the parameters $\mathbf{w}_k$, $b_k$, and $\mathbf{c}_k$, and VLAD with the parameter $\mathbf{c}_k$. This makes SuperVLAD not affected by the cluster center $\mathbf{c}_k$ and provides greater flexibility compared to NetVLAD. Besides, the number of parameters in SuperVLAD is about half less than that in NetVLAD. It is worth noting that Eq. 4 is different from applying the attention mechanism to local features. Our approach involves assigning features to clusters (rather than applying attention to features). More specifically, given a feature, the sum of the weights of assigning it to all clusters is the constant 1. However, for a given cluster, the sum of the weights of assigning all features to it is not fixed (for attention, it is the constant 1). The recent work SALAD [27] also sums the local features with the soft-assignment weights to get the global descriptor. However, its soft assignment relies on clusters (the so-called optimal transport between clusters and local features), *i.e.*, not free from clusters, and requires iterative computations to solve. Its parameters are no less than NetVLAD. These are different from our lightweight and low-computation SuperVLAD. Besides, In SuperVLAD we set only a small number of clusters, namely 4, which makes the output compact without additional dimensionality reduction techniques. As shown in Fig. 2, the main module of SuperVLAD is a 1×1 convolution (conv) and softmax for soft-assignment, and a VLAD core for aggregation. Unlike NetVLAD, the VLAD core of SuperVLAD has no parameters, and all trainable parameters exist only in the conv layer. The matrix $V$ is finally intra-normalized, flattened into a vector, and entirely L2-normalized as the output global descriptor.

Moreover, we also absorb GhostVLAD[65], which extends NetVLAD by introducing "ghost" clusters. Specifically, it adds additional $G$ ghost clusters that are used for the soft assignment in the same way as the original $K$ clusters, but the ghost clusters are disregarded in the aggregation process and do not directly contribute to the final output. Ghost clusters can be used to correspond to useless objects in VPR, such as sky, ground, dynamic objects, and so forth. In SuperVLAD, the number of ghost clusters is set to only one (*i.e.*, $G = 1$).

### 3.3 1-Cluster VLAD

Due to the high dimensionality of the descriptors output by NetVLAD, some SOTA VPR methods use learnable linear projections or GeM pooling for low-dimensional representations, *e.g.*, CosPlace [7], EigenPlaces [10], and MixVPR [2]. The GeM pooling is particularly popular because it easily produces global representations with the same dimensionality as the pooled local/patch features, although it may be lacking in performance. Here, we design a 1-cluster VLAD algorithm that can produce very low dimensional descriptors, which has the same output dimensionality as GeM but performs better. Specifically, we only need to set the number of useful clusters in SuperVLAD to 1, and the number of ghost clusters to greater than 1, (we set it to 2), *i.e.*, $K$=1 and $G$=2, to get the 1-cluster VLAD. This means that all descriptors representing objects relevant to VPR are assigned to the same cluster, and their weighted sum (the weight is $a_1(\mathbf{x}_i)$) is used as the final global descriptor.

---

[1] In implementation, $b_k$ can also be removed (i.e., $b_k$=0), as in the different implementation versions of NetVLAD (refer to https://github.com/Relja/netvlad/blob/master/README_more.md#netvlad-versions).

### 3.4 Cluster-wise Cross-image Interaction

This subsection introduces an optional process for SuperVLAD (not for 1-Cluster VLAD). To further enhance the performance, we draw inspiration from the cross-image encoder proposed in CricaVPR [39], which can use the cross-image variations as a cue to guide the representation learning and produce more robust image features through cross-image interaction in a batch. However, directly using the

Table 1: Summary of the evaluation datasets.

| Dataset | Description | Number | |
|---|---|---|---|
| | | Database | Queries |
| Pitts30k-test | urban, panorama | 10,000 | 6816 |
| MSLS-val | urban, suburban | 18,871 | 740 |
| Nordland | natural, seasonal | 27,592 | 27,592 |
| SPED-test | various scenes | 607 | 607 |

transformer encoder to process the entire SuperVLAD vector would incur significant memory and running time overhead, proportional to the square of the vector size. To overcome this issue, we first split the SuperVLAD descriptors by clusters and then input them into the cross-image encoder ( *i.e.*, two stacked transformer encoders) for cluster-wise cross-image interaction. That is, this encoder will model the correlation between the aggregated features belonging to the same cluster of all images in a batch. We then concatenate the aggregated features belonging to different clusters of an image and L2 normalize again to get the final descriptor of this image.

## 4 Experiments

### 4.1 Datasets and Performance Evaluation

We conduct experiments on several VPR benchmark datasets, which cover various challenges in VPR such as viewpoint changes, condition changes, and perceptual aliasing. Table 1 provides a concise summary of them. **Pitts30k** [54] mainly exhibits severe viewpoint changes in the urban environment. **MSLS** [57] is a comprehensive dataset comprising images collected in urban, suburban, and natural scenes over 7 years, and encompasses a wide range of visual changes (viewpoint and condition changes). **Nordland** [10] exhibits seasonal changes in natural and suburban scenes. **SPED** [11] consists of low-quality and high-scene-depth images captured from diverse scenes with various condition changes. More details are in Appendix G.

In our experiments, we employ the Recall@N (*i.e.*, R@N) to assess the recognition performance of VPR methods. This metric quantifies the percentage of queries for which at least one of the N retrieved reference images falls within a predefined threshold of the ground truth. We set the threshold to 25 meters for Pitts30k and MSLS, ±10 frames for Nordland, unique counterpart for SPED, following common evaluation procedures [54, 57, 10, 11].

### 4.2 Implementation Details

Here we describe the implementation details of training the DINOv2-based SuperVLAD model on the GSV-Cities [1] dataset for SOTA comparison. More details of training the other transformer-based models for the ablation study are in Appendix E. We implement our experiments on two NVIDIA GeForce RTX 3090 GPUs using PyTorch. The size of the input image is 224×224 in training (322×322 in inference) and the token dimension of the DINOv2-base backbone is 768. The descriptor dimension of SuperVLAD is 3072 and that of 1-cluster VLAD is 768. We only fine-tune the last four transformer encoder layers (freeze the previous layers) of the DINOv2 backbone. For the loss function, we utilize the multi-similarity loss [56] and set its hyperparameters as in [1]. The model training is performed using the Adam optimizer with an initial learning rate of 0.00005, halved every 3 epochs. Considering that the cross-image encoder is not initialized, we use a larger learning rate (0.0001) to train it separately. Each training batch consists of 120 places, with 4 images per place, resulting in a total of 480 images. An inference batch consists of 8 images (except for the SPED dataset where the batch size is 4). The training process is terminated when the performance on MSLS-val does not improve for three epochs. The actual number of effective training epochs is 7, and the training time is 81.6 minutes.

### 4.3 Comparisons with State-of-the-Art Methods

In this section, we compare the proposed SuperVLAD with several excellent VPR methods, including seven one-stage methods using global feature retrieval (like ours): NetVLAD [4], SFRS [21],

Table 2: Comparison to state-of-the-art methods on four VPR benchmark datasets. The best results are highlighted in **bold** and the second are underlined. The descriptor dimensionalities of two-stage methods are not displayed.

| Method | Backbone | Dim | Pitts30k | | | MSLS-val | | | Nordland | | | SPED | | |
|---|---|---|---|---|---|---|---|---|---|---|---|---|---|---|
| | | | R@1 | R@5 | R@10 | R@1 | R@5 | R@10 | R@1 | R@5 | R@10 | R@1 | R@5 | R@10 |
| NetVLAD [4] | VGG16 | 32768 | 81.9 | 91.2 | 93.7 | 53.1 | 66.5 | 71.1 | 6.4 | 10.1 | 12.5 | 70.2 | 84.5 | 89.5 |
| SFRS [21] | VGG16 | 4096 | 89.4 | 94.7 | 95.9 | 69.2 | 80.3 | 83.1 | 16.1 | 23.9 | 28.4 | 80.2 | 92.6 | 95.4 |
| TransVPR [55] | / | / | 89.0 | 94.9 | 96.2 | 86.8 | 91.2 | 92.4 | 63.5 | 68.5 | 70.2 | 85.7 | 90.9 | 91.8 |
| CosPlace [7] | VGG16 | 512 | 88.4 | 94.5 | 95.7 | 82.8 | 89.7 | 92.0 | 58.5 | 73.7 | 79.4 | 75.5 | 87.0 | 89.6 |
| MixVPR [2] | ResNet50 | 4096 | 91.5 | 95.5 | 96.3 | 88.0 | 92.7 | 94.6 | 76.2 | 86.9 | 90.3 | 84.7 | 92.3 | 94.4 |
| EigenPlaces [10] | ResNet50 | 2048 | 92.5 | 96.8 | 97.6 | 89.1 | 93.8 | 95.0 | 71.2 | 83.8 | 88.1 | 70.2 | 83.5 | 87.5 |
| SelaVPR [41] | DINOv2-L | / | 92.8 | 96.8 | 97.7 | 90.8 | 96.4 | 97.2 | 87.3 | 93.8 | 95.6 | 89.5 | 94.6 | 95.9 |
| CricaVPR [39] | DINOv2-B | 4096 | 94.9 | 97.3 | **98.2** | 90.0 | 95.4 | 96.4 | 90.7 | 96.3 | 97.6 | 91.4 | 95.2 | 96.2 |
| SALAD [27] | DINOv2-B | 8448 | 92.5 | 96.4 | 97.5 | **92.2** | 96.4 | 97.0 | 89.7 | 95.5 | 97.0 | 92.1 | 96.2 | 96.5 |
| SuperVLAD | DINOv2-B | 3072 | **95.0** | **97.4** | **98.2** | **92.2** | **96.6** | **97.4** | **91.0** | **96.4** | **97.7** | **93.2** | **97.0** | **98.0** |

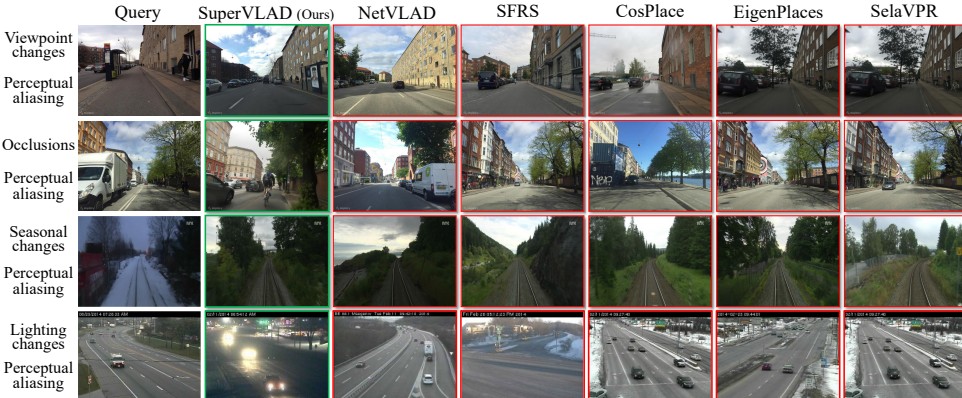

Figure 4: Qualitative results. In these four challenging examples (covering viewpoint variations, condition variations, dynamic objects, etc.), our SuperVLAD successfully retrieves the right database images, while other methods get the wrong results.

CosPlace [7], MixVPR [2], Eigenplaces [10], CricaVPR [39], and SALAD [27], as well as two two-stage methods with re-ranking: TransVPR [55] and SelaVPR [41]. Note that MixVPR, CricaVPR, SALAD, and our SuperVLAD use the same training dataset, *i.e.*, GSV-Cities. Meanwhile, CosPlace and EigenPlaces are trained on the purpose-built extra large-scale (SF-XL) datasets. Additionally, the latest works SelaVPR, CricaVPR, and SALAD all use the foundation model DINOv2 as the backbone (SelaVPR using DINOv2-large and others using DINOv2-base) and achieve the SOTA performance on the VPR task. So here we follow them and use the DINOv2-base backbone. The details of the above methods can be seen in Appendix H. Table 2 shows the quantitative results and our SuperVLAD achieves the best results on all these datasets.

The methods based on the DINOv2 backbone, including SelaVPR, CricaVPR, SALAD, and our SuperVLAD, all achieve excellent performance and outperform the remainder methods on these datasets with diverse challenges. These methods all fine-tune DINOv2 in different ways, which shows that based on the powerful feature representation capability of DINOv2, coupled with appropriate fine-tuning, it is sufficient to cope with most challenges in the VPR task. However, our method uses a more compact feature representation (than CricaVPR and SALAD) and outperforms other methods on the four datasets. In particular, it achieves 93.2% R@1 on the SPED dataset, demonstrating significant advantages compared to other global-retrieval-based methods and two-stage methods. This indicates the high robustness of our method to handle condition variations in datasets containing low-quality and high-scene-depth images. Additionally, both CricaVPR and SuperVLAD use the cross-image encoder and thus achieve significantly better performance than other methods on Pitts30k and Nordland, which are able to provide different images of the same place in a batch and mutually improve condition invariance and viewpoint invariance [39]. However, such benefit cannot be obtained on MSLS-val and SPED (all query images from different places), our approach still outperforms all other methods (and is obviously better than CricaVPR).

Fig. 4 qualitatively shows the superior performance of our SuperVLAD in some difficult scenes. These examples exhibit various challenges, such as drastic condition changes, viewpoint changes, and occlusions (caused by dynamic objects). Other methods mostly return similar images from different places, *i.e.*, suffer from perceptual aliasing and fail to get the right results. However, our SuperVLAD retrieves the correct reference images, showing high robustness against these challenges.

Fig. 5 simultaneously shows the R@1 on Pitts30k, the inference time of a single image, and the descriptor dimensionality. Among the four methods (SFRS, MixVPR, SALAD, and SuperVLAD), MixVPR uses the CNN backbone and the feature mixing method to get global descriptors, which achieves the shortest inference time. The remaining three methods use VLAD-

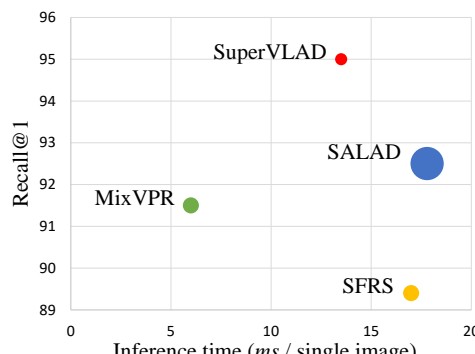

Figure 5: The comparison of some global-retrieval-based methods in Recall@1 (on Pitts30k), inference time (ms/single image), and descriptor dimensionality. The diameter of each dot is proportional to the descriptor dimension. Our SuperVLAD gets the best R@1 with the most compact descriptor.

related methods to get global features, and our SuperVLAD achieves the fastest inference speed among them. Although SFRS is based on the CNN model, it uses NetVLAD with 64 clusters to obtain high-dimensional image features and then uses PCA for dimensionality reduction (time-consuming). SALAD and our SuperVLAD are based on the foundation model DINOv2. However, SALAD uses the Sinkhorn algorithm to iteratively compute optimal transport assignment between clusters and local features, which takes more inference time than our method (17.8ms vs. 13.5ms). Besides, the number of parameters in our SuperVLAD aggregator is much lower than that of SALAD (less than 3/1000), as shown in Table 3. Our SuperVLAD uses a lightweight and low-compute aggregation layer to get the compact global descriptor (without extra dimensionality reduction), which simultaneously has fast inference speed, low-dimensional descriptors, and excellent recognition performance.

## 4.4 Ablation Study

In this section, we conduct a series of ablation experiments to demonstrate the effectiveness of the proposed SuperVLAD and 1-Cluster VLAD. The cross-image encoder is not used by default in ablation experiments. The SuperVLAD using the cross-image encoder is denoted as SuperVLAD[†].

**Effect of SuperVLAD.** We first validate the effectiveness of SuperVLAD by comparing

Table 3: The number of parameters of SALAD and SuperVLAD that both use the DINOv2-base backbone. The value in parentheses is the number of parameters in the optional cross-image encoder.

| Method | Total (M) | Trainable (M) | Aggregator (M) |
|---|---|---|---|
| SALAD | 88.0 | 29.8 | 1.4 |
| SuperVLAD | 86.6 (+11.0) | 28.4 (+11.0) | 0.0038 |

it to NetVLAD. To be fair, we do not use the cross-image encoder in SuperVLAD, and apply multiple different transformer-based models as the backbone, including the ViT [17] and CCT [23] models pre-trained on ImageNet [15], and the foundation model DINOv2 [46] (that is also a ViT model) pre-trained on the large-scale curated dataset. We conduct experiments with three training sets (MSLS, Pitts30k, and GSV-Cities) and two test sets (MSLS and Pitts30k). To validate that SuperVLAD has superior generalization ability over NetVLAD in addressing the domain shift issue mentioned above, we focus on the cross-domain inference performance of models. For example, the performance of a model trained on Pitts30k when tested on MSLS, or vice versa. The results are shown in Table 4. Note that Pitts30k contains only urban scene images, while MSLS covers urban, suburban, and natural scenes, and GSV-Cities is a large-scale training set with accurate supervision and diverse visual variations. Thus, the performance of a model trained on Pitts30k and tested on MSLS best reflects domain generalization, while the reverse is less indicative. The model trained on GSV-Cities encounters almost no domain drift issues on these test sets (Pitts30k and MSLS-val). The experimental results largely align with these facts and the above theory. Training the SuperVLAD model (based on CCT and DINOv2) on Pitts30k and testing it on Pitts30k yields similar results to NetVLAD. However, when tested on MSLS, it shows an absolute R@1 improvement of 6.8% (with CCT) and 2.4% (with DINOv2), respectively. The SuperVLAD model trained on MSLS using ViT

Table 4: Comparison of NetVLAD and SuperVLAD with different backbones and training sets. Since the token dimension of CCT is only half of ViT, models based on it have 8 clusters, while models based on ViT/DINOv2 have 4 clusters. The values in parentheses indicate the change in results of SuperVLAD relative to NetVLAD: red for increase, green for decrease, and black for no change.

| Method | Backbone | Training set | Pitts30k | | | MSLS-val | | |
|---|---|---|---|---|---|---|---|---|
| | | | R@1 | R@5 | R@10 | R@1 | R@5 | R@10 |
| NetVLAD | ViT | MSLS | 79.0 | 90.9 | 94.1 | 75.7 | 89.5 | 91.8 |
| SuperVLAD | | | 79.4(+0.4) | 91.9(+1.0) | 94.7(+0.6) | 75.9(+0.2) | 88.8(-0.7) | 91.8(+0.0) |
| NetVLAD | CCT | Pitts30k | 84.7 | 93.0 | 95.3 | 55.4 | 68.5 | 73.1 |
| SuperVLAD | | | 84.7(+0.0) | 93.0(+0.0) | 95.2(-0.1) | 62.2(+6.8) | 73.1(+4.6) | 77.3(+4.2) |
| NetVLAD | DINOv2 | Pitts30k | 89.4 | 95.7 | 96.9 | 72.7 | 84.7 | 87.2 |
| SuperVLAD | | | 89.4(+0.0) | 95.8(+0.1) | 97.1(+0.2) | 75.1(+2.4) | 84.6(-0.1) | 87.4(+0.2) |
| NetVLAD | DINOv2 | GSV-Cities | 92.3 | 96.8 | 97.7 | 91.6 | 96.2 | 96.8 |
| SuperVLAD | | | 92.6(+0.3) | 96.4(-0.4) | 97.5(-0.2) | 92.2(+0.6) | 95.9(-0.3) | 96.8(+0.0) |

Table 5: Comparison of SuperVLAD with and without the ghost cluster. "SV" is short for SuperVLAD. The methods with the "-ng" suffix are those without the ghost cluster. Specifically, DINOv2-SV is the model based on DINOv2 and trained on GSV-Cities as detailed in Table 4.

| Method | Pitts30k | | | MSLS-val | | |
|---|---|---|---|---|---|---|
| | R@1 | R@5 | R@10 | R@1 | R@5 | R@10 |
| CCT-SV-ng | 84.1 | 92.9 | 95.1 | 60.3 | 72.2 | 76.9 |
| CCT-SV | **84.7** | **93.0** | **95.2** | **62.2** | **73.1** | **77.3** |
| DINOv2-SV-ng | 92.4 | **96.5** | **97.6** | **92.4** | **96.4** | **96.9** |
| DINOv2-SV | **92.6** | 96.4 | 97.5 | 92.2 | 95.9 | 96.8 |

Table 6: Comparison of the very low-dimensional global descriptors with the same dimensions as the local descriptors. That is, all methods produce 768-dim global descriptors (using DINOv2-base backbone). All models are trained on GSV-Cities.

| Method | Pitts30k | | | MSLS-val | | |
|---|---|---|---|---|---|---|
| | R@1 | R@5 | R@10 | R@1 | R@5 | R@10 |
| GeM Pooling | 89.5 | 95.0 | 96.3 | 85.4 | 93.0 | 94.3 |
| class token | 91.4 | **96.2** | **97.4** | 88.4 | 95.1 | **96.4** |
| 1-ClusterVLAD | **91.6** | **96.2** | **97.4** | **90.4** | **95.3** | **96.4** |

exhibits similar test results to NetVLAD on MSLS-val but still has a slight advantage when tested on Pitts30k. These results demonstrate that our SuperVLAD has better domain generalization than NetVLAD. Since the GSV-Cities dataset covers the various visual variations in VPR and provides accurate supervision for fine-tuning, both models trained on it perform very well on Pitts30k and MSLS-val. However, this does not mean that our SuperVLAD has no advantage over NetVLAD with such a large-scale VPR training set. SuperVLAD achieves the same performance as NetVLAD with fewer parameters and computations, *e.g.*, offering equivalent performance at a lower cost.

**Effect of ghost clusters** and **the performance of 1-Cluster VLAD.** To investigate the impact of ghost clusters, we conduct two sets of experiments based on CCT and DINOv2 to compare the performance with and without ghost clusters. All models also do not use the cross-image encoder. The CCT-based model is trained on Pitts30k and the DINOv2-based model is trained on GSV-Cities. The results are shown in Table 5. When based on DINOv2, whether using ghost clusters or not achieves excellent performance, with little to no difference in their performance (without using ghost clusters even slightly better). However, based on CCT and trained on Pitts30k, the use of ghost clusters resulted in a certain improvement. This indicates that the ghost cluster is still necessary. More importantly, we leverage it to achieve 1-Cluster VLAD. We compared the proposed 1-Cluster VLAD with two commonly used methods (GeM pooling and class token) that can output very compact features of the same dimension as local features (as does our 1-Cluster VLAD). The results, as shown in Table 6, demonstrate the obvious superiority of 1-Cluster VLAD compared to the other two methods. So, our 1-Cluster VLAD can be used as a new choice for very low-dimensional descriptors.

**Effect of the number of clusters.** To investigate the impact of the number of clusters on global descriptors, we conducted two sets of experiments using SuperVLAD with and without the cross-image encoder. All models utilize DINOv2 as the backbone and are trained on GSV-Cities. Results (see Table 7) indicate that even with a small number of clusters, there is no significant performance decrease. This is primarily attributed to the powerful feature representation capability of the transformer-based foundation model DINOv2, while the large-scale VPR training dataset GSV-Cities offers appropriate supervision for fine-tuning. Even with a small number of clusters, it allows for reasonable and effective classification/clustering of objects that are relevant to place recognition (*e.g.*, buildings, vegetation). Unlike most VLAD-related methods that use dozens of clusters, our experiments show

that just 4 clusters can achieve robust VPR in most cases, which provides compact global descriptors without the need for additional dimensionality reduction techniques.

**Effect of the cross-image encoder.** Table 7 also presents the comparison of SuperVLAD with and without cross-image encoder (*i.e.*, our cluster-wise cross-image interaction). It is evident that using a cross-image encoder consistently leads to performance improvements. However, the performance enhancement on Pitts30k is significantly greater than that on MSLS. This is because Pitts30k can provide a batch of images with different viewpoints from the same place during inference, which allows different images to directly improve the ro-

Table 7: Comparison of the SuperVLAD ablated versions with different numbers of clusters, as well as with and without the cross-image encoder (the former denoted as SuperVLAD$^{\dagger}$).

| Method | No. Clusters | Pitts30k | | | MSLS-val | | |
|---|---|---|---|---|---|---|---|
| | | R@1 | R@5 | R@10 | R@1 | R@5 | R@10 |
| SuperVLAD | 4 | **92.6** | 96.4 | 97.5 | **92.2** | 95.9 | **96.8** |
| | 16 | 92.4 | **96.7** | **97.7** | 90.5 | **96.2** | **96.8** |
| | 64 | 92.5 | 96.5 | **97.7** | 91.4 | 95.8 | 96.5 |
| SuperVLAD$^{\dagger}$ | 4 | **95.0** | 97.4 | **98.2** | 92.2 | 96.6 | **97.4** |
| | 16 | 93.6 | 96.9 | 97.9 | 91.4 | 96.2 | 97.2 |
| | 64 | 94.6 | **97.5** | 98.1 | **92.3** | **96.8** | 97.3 |

bustness of each other through the cross-image interaction, while MSLS cannot. Considering that conditions like the Pitts30k dataset might not be available in practical application, we consider the cross-image encoder as an optional component for performance enhancement. It is worth noting that, even without it, our method still achieves performance comparable to the SOTA methods.

# 5  Conclusions

In this paper, we introduced SuperVLAD, a compact and robust global descriptor for VPR. By eliminating the need for cluster centers and utilizing a small number of clusters, SuperVLAD achieved improved domain generalization and produced more compact descriptors. We also designed a 1-cluster VLAD descriptor with extremely low dimensionality by introducing supernumerary ghost clusters during the soft assignment. Experimental results on various transformer-based backbones validated the effectiveness of SuperVLAD, producing more robust features than NetVLAD with a lighter aggregation layer. Furthermore, the DINOv2-based SuperVLAD also outperformed SOTA methods on several VPR benchmark datasets with more compact global descriptors.

**Limitations & Future Work.** While our study presents some improvements in VPR, particularly in enhancing domain generalization ability, reducing the dimensionality of global descriptors, and reducing the number of parameters for the aggregation layer, we acknowledge two limitations of our work: **Firstly**, the current success of SuperVLAD relies on the use of the transformer backbone. Although our experimental results provided in Appendix C demonstrate that SuperVLAD still has certain advantages over NetVLAD when using a CNN backbone with a small number of clusters, it is not as good as NetVLAD in more commonly used settings. This can be seen as a limitation of SuperVLAD. **Secondly**, our method uses the cross-image encoder to further enhance performance, similar to CricaVPR [39]. It is necessary to set the inference batch size to an appropriate value. Setting it to 1 directly renders the cross-image encoder ineffective during inference, resulting in a performance drop due to the gap between training and inference. Since we can only use a single-frame query in some practical applications, we compromise to treat the cross-image encoder as an optional module that is not used (during both training and inference) when multiple-frame inference is not feasible. Even without the cross-image encoder, our SuperVLAD can achieve good performance. Some additional discussion on the limitations of SuperVLAD is in Appendix B. In future work, we will try to address the above limitations, and further assess the long-term stability and potential performance changes of the model, which are important for practical deployments.

## Acknowledgments and Disclosure of Funding

This work was supported by the National Key R&D Program of China (2022YFB4701400/4701402), SSTIC Grant (KJZD20230923115106012, KJZD20230923114916032, GJHZ20240218113604008), Beijing Key Lab of Networked Multimedia, the Project of Pengcheng Laboratory (PCL2023A08, PCL2024Y02), and National Natural Science Foundation of China (62402252).

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

# A    Broader Impacts

Our research on VPR, particularly the development of the SuperVLAD and 1-Cluster VLAD methods, has been motivated by the potential to significantly enhance applications in augmented reality, robot localization, and autonomous driving. The potential benefits and disadvantages depend on how people choose to use our method and how they handle failure cases.

**Positive Impacts on Technology Advancement.** The introduction of our novel method offers a more compact and generalizable feature representation, which could lead to more efficient and robust retrieval capability for downstream applications.

**Potential Failure Cases.** Although our method has shown high performance on evaluation metrics, there remains a small probability of failure cases, which could potentially lead people or autopilots astray. Nevertheless, suppose our method is used in conjunction with other methods and information for autonomous driving systems, then our method mostly introduces additional safety and efficiency to the overall system.

**Potential Malicious Uses.** Although our research on better feature representation is foundational and not tied to particular applications, the VPR system may be exploited for invasive surveillance or social media monitoring, leading to privacy concerns. It is important to avoid unethical use of VPR studies for negative purposes.

# B    More Comparisons between NetVLAD and SuperVLAD

We have detailed the benefits of removing cluster centers in SuperVLAD in the main paper. This section provides more comparisons and analyses between NetVLAD and SuperVLAD, mainly analyzing the possible negative effects of removing cluster centers (i.e., the advantages of using cluster centers in NetVLAD) to illustrate the limitations (or characteristics) of SuperVLAD.

First, the cluster centers have the potential to play a guiding role in improving the accuracy of assigning local features to different clusters. Especially when local features are assigned to fine-grained clusters (*i.e.*, a large number of clusters), this strength of NetVLAD over SuperVLAD will gradually become apparent. Although the DINOv2-based NetVLAD model with a small number of clusters can achieve performance comparable to a large number of clusters when training on the large-scale dataset GSV-Cities, it still makes sense to set a large number of clusters for a simpler backbone and a smaller training dataset. Here we conduct complementary experiments using the CCT backbone and Pitts30k training dataset. The results are shown in Table 8. In this case, SuperVLAD has a clear advantage over NetVLAD with a small number of clusters, but this advantage gradually disappears with a large number of clusters. That is, NetVLAD shows a more pronounced improvement with a larger number of clusters. However, when using a large number of clusters, the memory occupied by the aggregation layer usually becomes significant in the training stage. The SuperVLAD can significantly reduce GPU memory usage because it has only half the trainable parameters of NetVLAD. Moreover, from another perspective, SuperVLAD experiences less performance degradation with fewer clusters, indicating its good ability to produce low-dimensional global features by directly reducing the number of clusters. Therefore, we do not consider this as a limitation of SuperVLAD but rather as one of its characteristics (*i.e.*, more suitable for a small number of clusters than NetVLAD).

Secondly, using the local features to subtract the cluster center to get the residual can more accurately capture the subtle differences of local features (improve the ability to distinguish details). However, when we use the transformer-based backbone (especially the foundation model) to provide powerful local features (*i.e.*, patch tokens), this can basically be ignored. So we use the transformer backbone for our SuperVLAD, which can be considered a limitation. We will discuss the use of CNN backbone in the next section.

# C    Additional Comparisons Using the CNN Backbone

Although we have mentioned in the main paper that our SuperVLAD requires a transformer-based backbone to extract local features (*i.e.*, patch tokens) of images, here we also present experimental results when applied to a CNN backbone (specifically VGG16) in order to observe the limitations of our method. Considering that previous works using NetVLAD to aggregate CNN feature maps

Table 8: Comparison of NetVLAD and SuperVLAD with different numbers of clusters using the CCT backbone and the Pitts30k training dataset. The smaller the number of clusters, the greater the advantage of SuperVLAD over NetVLAD. The values in parentheses show the change in results of SuperVLAD relative to NetVLAD: red for increase, green for decrease, and black for no change.

| Method | No. Cluster | Pitts30k | | | MSLS-val | | |
|---|---|---|---|---|---|---|---|
| | | R@1 | R@5 | R@10 | R@1 | R@5 | R@10 |
| NetVLAD | 2 | 78.3 | 90.5 | 93.5 | 42.4 | 53.4 | 57.2 |
| SuperVLAD | | 80.1(+1.8) | 90.9(+0.4) | 93.9(+0.4) | 48.0(+5.6) | 61.9(+8.5) | 65.5(+8.3) |
| NetVLAD | 8 | 84.7 | 93.0 | 95.3 | 55.4 | 68.5 | 73.1 |
| SuperVLAD | | 84.7(+0.0) | 93.0(+0.0) | 95.2(-0.1) | 62.2(+6.8) | 73.1(+4.6) | 77.3(+4.2) |
| NetVLAD | 64 | 85.4 | 93.3 | 95.3 | 62.0 | 72.6 | 77.2 |
| SuperVLAD | | 85.1(-0.3) | 93.1(-0.2) | 95.4(+0.1) | 63.0(+1.0) | 73.5(+0.9) | 77.7(+0.5) |

Table 9: Comparison of NetVLAD and SuperVLAD using the CNN backbone (VGG16) with different numbers of clusters. We also provide the results of the corresponding DINOv2-based models as a reference. All models are trained on GSV-Cities.

| Method | Backbone | No. Cluster | Pitts30k | | | MSLS-val | | |
|---|---|---|---|---|---|---|---|---|
| | | | R@1 | R@5 | R@10 | R@1 | R@5 | R@10 |
| NetVLAD | DINOv2 | 4 | 92.3 | 96.8 | 97.7 | 91.6 | 96.2 | 96.8 |
| SuperVLAD | | | 92.6(+0.3) | 96.4(-0.4) | 97.5(-0.2) | 92.2(+0.6) | 95.9(-0.3) | 96.8(+0.0) |
| NetVLAD | DINOv2 | 64 | 91.9 | 96.1 | 97.3 | 92.3 | 96.2 | 96.6 |
| SuperVLAD | | | 92.5(+0.6) | 96.5(+0.4) | 97.7(+0.4) | 91.4(-0.9) | 95.8(-0.4) | 96.5(-0.1) |
| NetVLAD | VGG16 | 4 | 83.8 | 91.9 | 94.7 | 73.5 | 84.1 | 87.0 |
| SuperVLAD | | | 84.5(+0.7) | 92.0(+0.1) | 94.4(-0.3) | 76.4(+2.9) | 85.7(+1.6) | 87.3(+0.3) |
| NetVLAD | VGG16 | 64 | 87.6 | 93.5 | 95.4 | 80.7 | 88.4 | 90.3 |
| SuperVLAD | | | 86.3(-1.3) | 93.0(-0.5) | 94.5(-0.9) | 78.8(-1.9) | 87.8(-0.6) | 90.3(+0.0) |

typically set a large number of clusters (*e.g.*, 64), we provide configurations with both 4 and 64 clusters. The experimental results are shown in Table 9. We also provide the results using DINOv2 as the backbone for reference. All models are trained on the GSV-Cities dataset. For the VGG16 backbone, it can be observed that with only 4 clusters, our SuperVLAD also has an advantage over NetVLAD. However, under the more commonly used setting of 64 clusters, SuperVLAD lags behind NetVLAD. Although there is no significant difference in performance between 4 and 64 clusters when using the DINOv2 backbone, both NetVLAD and SuperVLAD perform significantly better with 64 clusters when using the CNN backbone. Considering that the CNN-based (NetVLAD and SuperVLAD) models with 4 clusters do not perform well even using the large-scale GSV-Cities dataset for training, *i.e.*, a large number of clusters is necessary in this case, we acknowledge that SuperVLAD is inferior to NetVLAD when using a CNN backbone. Therefore, we recommend its use primarily with the transformer backbone (especially foundation models to provide powerful patch features).

# D   Comparisons on More Datasets

This section provides comparisons with SOTA methods (SelaVPR [41] and SALAD [27]) on more datasets, as shown in Table 10. Pitts250k and Tokyo247 are two datasets with large databases (more than 75k images), on which our method outperforms other methods. Baidu Mall [49, 30] is the only indoor dataset, so there is a clear gap between it and other datasets. Our work also achieves good results on this dataset. However, it should be noted that this is not the domain gap that we wanted to solve in our work. This is because the objects that need to be paid attention to in indoor images are significantly different from those in outdoor images. Although different outdoor datasets may have domain gaps due to scenes (*e.g.*, urban v.s. suburban) or conditions (*i.e.*, illumination), the objects that need to be paid attention to are basically always buildings and vegetation, which is not true for indoor datasets (see Fig. 6 for some examples). Therefore, we think that the issue of indoor VPR should be solved from other insights, rather than the idea of this work.

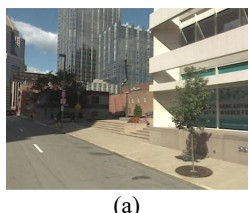 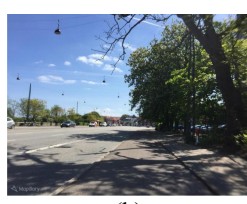 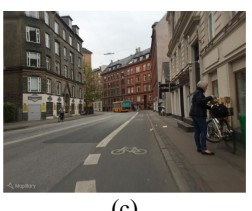 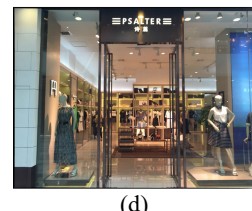

| (a) | (b) | (c) | (d) |

Figure 6: The example of data gap between different datasets (Pitts30k, MSLS and Baidu Mall). Pitts30k is a dataset consisting of urban scene images, as shown in (a). MSLS contains urban and suburban (even natural) scene images, as shown in (b) and (c). There is an obvious domain gap between these two datasets, which is the issue our method attempts to address. Besides, (d) shows an example from the Baidu Mall indoor dataset. There is another gap between indoor data and outdoor data, which is mainly caused by the fact that the objects that need to be paid attention to in the two types of data are usually different.

Table 10: Comparison to SOTA methods (SelaVPR and SALAD) on additional datasets. The inference batch size is 8 for Pitts250k and Tokyo247, and 4 for Baidu Mall.

| Method | Pitts250k | | | Tokyo247 | | | Baidu Mall | | |
|---|---|---|---|---|---|---|---|---|---|
| | R@1 | R@5 | R@10 | R@1 | R@5 | R@10 | R@1 | R@5 | R@10 |
| SelaVPR [41] | 95.7 | 98.8 | 99.2 | 94.0 | 96.8 | 97.5 | 66.6 | 79.6 | 85.9 |
| SALAD [27] | 95.1 | 98.5 | 99.1 | 94.6 | 97.5 | 97.8 | 68.3 | **82.1** | 86.8 |
| SuperVLAD | **97.2** | **99.4** | **99.7** | **95.2** | **97.8** | **98.1** | **69.4** | 81.9 | **88.0** |

## E    Additional Implementation Details

In the main paper, we have described the implementation details of training DINOv2-SuperVLAD models on GSV-Cities for comparison with the SOTA methods. Here we introduce the details of the implementation of the ablation experiment. For the training of the models based on ViT and CCT backbones (and trained on MSLS/Pitts30k), we follow some previous work [9, 38]. The last two transformer encoder layers of the ViT backbone (ViT-Base) are truncated and the size of the input image is 224×224. The last four transformer encoder layers of the CCT backbone (CCT-14) are truncated and the layers before the third layer are frozen, using the input images with 384×384 pixels. For the training of VGG16-based models on GSV-Cities, the implementation is basically the same as the DINOv2-based model. However, the layers before the last block of the VGG16 backbone are frozen, as in [9]. ViT, CCT, and VGG16 are all pre-trained in ImageNet [15]. Following the visual geo-localization benchmark [9], we use the triplet loss to train models on Pitts30k and MSLS. Ten hard negative images are used in each triplet. The model training is performed using the Adam optimizer with a learning rate of 0.00001. The batch size is set to 4. The training process on MSLS is terminated when the result does not improve for 6 epochs (40000 images/epoch), and that on Pitts30k are 3 epochs (5000 images/epoch).

## F    Additional Qualitative Results

Fig. 4 in the main paper has presented a few qualitative results. As a supplement, Fig. 7, Fig. 8], Fig. 9, and Fig. 10 provide additional qualitative results for MSLS, Pitts30k, Nordland, and SPED. Examples from these datasets highlight challenging scenarios including severe condition changes, viewpoint variations, dynamic interference, few or no landmarks, etc. Ours SuperVLAD achieves correct results, while others fail. Moreover, we also provide failure cases of our approach as shown in Fig. 11, which suggests that future improvements of our method may need to consider how to retrieve images that are closer to the query image and improve robustness in natural scenes.

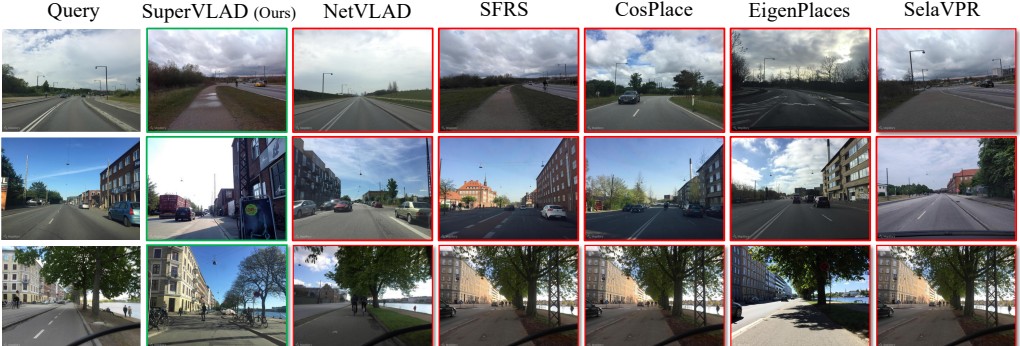

Figure 7: Qualitative results on the MSLS dataset. The first query lacks distinctive landmarks. The first two examples show significant viewpoint changes between the query and the correct reference image. For the third query, most of the other methods returned the same erroneous place due to the inability to distinguish small-scale differences in the building surface. Only our method provides the correct result.

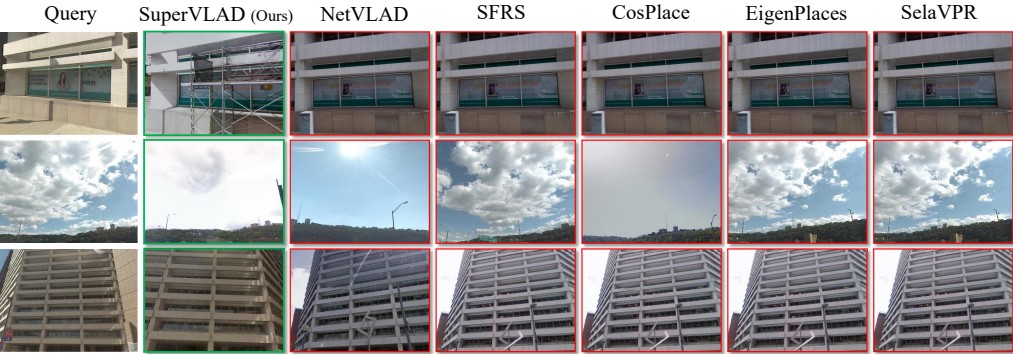

Figure 8: Qualitative results on the Pitts30k dataset. In the second example, Although most of the image area in the query image is the sky, our SuperVLAD accurately returns the correct image using only a small useful image area, whereas other methods return incorrect images. In the first and third examples, other methods retrieve images very similar to the query (still wrong), but our method can distinguish the different places through subtle differences, yielding the correct image.

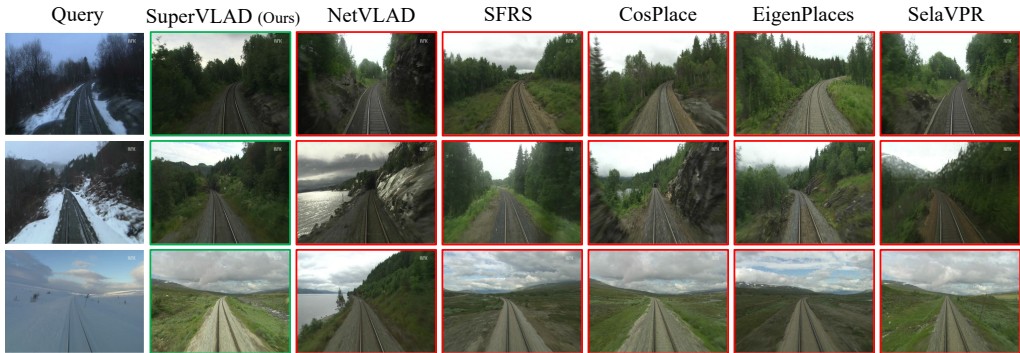

Figure 9: Qualitative results on the Nordland dataset. These three queries contain no discriminative landmarks and are subject to significant seasonal and lighting variations. The third example is particularly challenging, as most of the image region is obscured by snow and clouds and there are few discernible features along the road. Only ours SuperVLAD correctly returns the matched image.

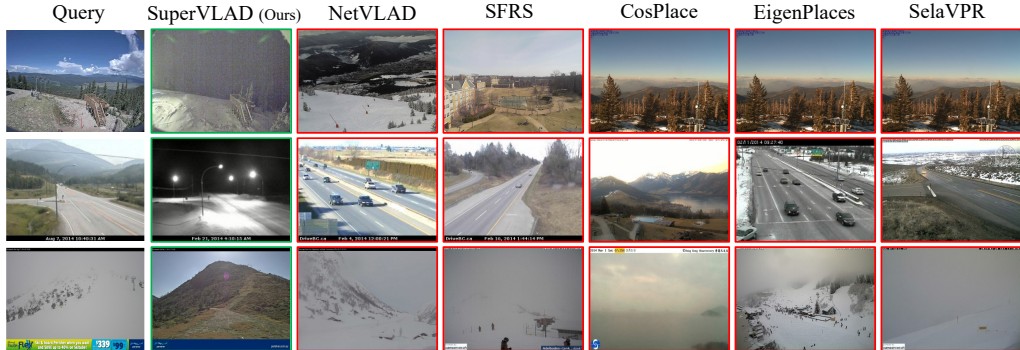

Figure 10: Qualitative results on the SPED dataset. In the second example, daytime query images mislead other methods into retrieving similar-looking daytime images from different places. Our method can return the right image at night, with significant visual changes. Due to low light, some discriminative objects are difficult to see clearly, and it is hard to recognize the places even by human experts. The third example shows a snowy query image (without discriminative landmarks), causing other methods to return snowy or foggy images. Only our method finds the correct image of the same place in different weather conditions.

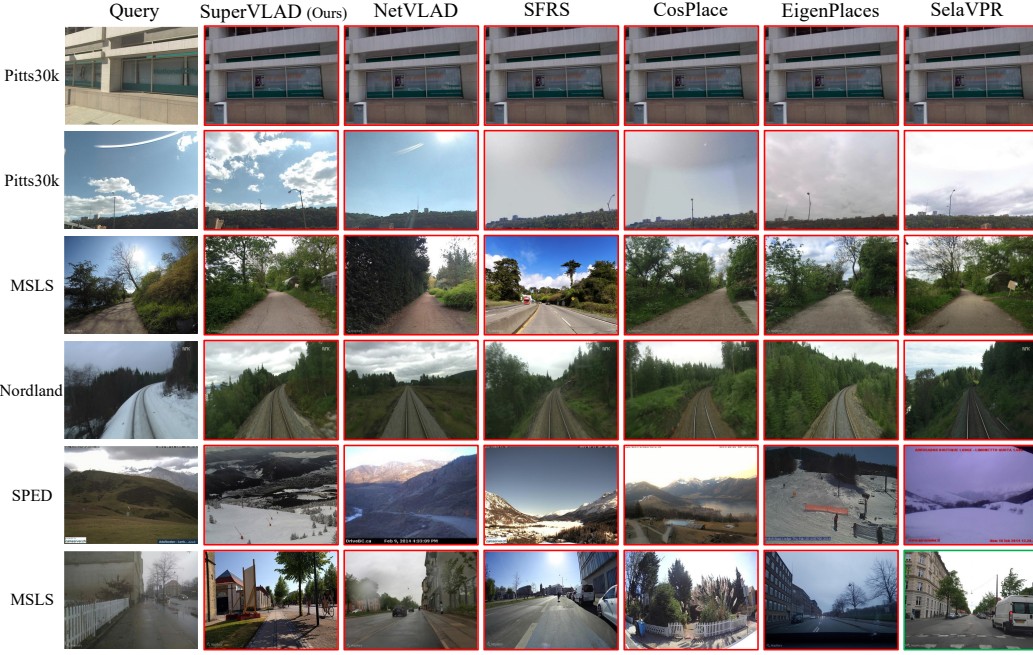

Figure 11: Failure Cases. For the first two examples (first two rows), our method (and some other methods) obtains database images that are geographically close to the query image but the radius exceeds the set threshold (25m). This phenomenon usually occurs when the camera is too close or too far away from the scene. For the third to fifth examples, all methods fail because the images are captured in natural scenes without landmarks, which seems to be a noteworthy challenge for current VPR methods. For the last challenging example (bad weather and large viewpoint changes), all global retrieval methods (including ours) fail, while SelaVPR based on local feature re-ranking returns the correct result. This shows that even though our method achieves good results using only global features, it is possible to further improve the results by local feature re-ranking.

## G   Dataset Details

**MSLS** [57]. The MSLS (Mapillary Street-Level Sequences) dataset is a large-scale VPR dataset with over 1.6 million images collected in urban and suburban scenes from 30 cities on six continents, provided with GPS coordinates and compass angles. It covers many causes of environmental changes, such as illumination, weather, season, and viewpoint changes, as well as dynamic objects. The dataset contains three sets: training, public validation (MSLS-val), and withheld test (MSLS-challenge). We evaluate models on the MSLS-val set. It should be noted that the MSLS val set used in the Visual Geo-localization Benchmark [9] and CosPlace [7] contains a different number of query images as the official version[2] [57]. To facilitate comparison with previous methods, we use the official version and run CosPlace on it, obtaining better results than those reported in [7].

**Pitts30k** [54]. The Pitts30k dataset is generated from Google Street View panoramas with GPS labels available. It provides 24 viewpoint images at each location, with large viewpoint variations as well as some dynamic objects. Pitts30k contains 10k database images in each of the training, validation, and test sets. Its test set is adopted to evaluate models in experiments.

**Nordland** [50]. The Nordland dataset is recorded from the same viewpoint in the front of a train in four seasons. So, there are severe condition (e.g., season and weather) changes but no viewpoint changes in this dataset. It mainly contains suburban and natural place images. The ground truth of this dataset is offered by the frame-level correspondence. Following previous works [10], we use the summer images as the database and the winter images as queries, each containing 27592 images.

**SPED** [11]. The SPED dataset is made up of low-quality and high-scene-depth images taken from CCTV cameras around the globe. The images in this dataset show various condition variations, such as lighting, weather, and seasonal changes. This dataset covers a diverse range of outdoor scenes, including forest landscapes, country roads, and urban environments. We use the SPED-test set for experiments, with 607 images in both its database and query set.

## H   Compared Methods Details

**NetVLAD** [4][3]. NetVLAD is a classic VPR method with a learnable VLAD layer that is pluggable into any CNN architecture. We use the PyTorch implementation with the VGG16 backbone trained on the Pitts30k dataset.

**SFRS** [21][4]. This work mines hard positive samples using self-supervised image-to-region similarities for training a more robust NetVLAD model. We use the official implementation with the model trained on Pitts30k in the comparison experiment.

**TransVPR** [55][5]. This work combines attentions from multiple levels of vision Transformer to yield global features for the candidate retrieval, then uses an attention mask to filter feature maps to get key-patch descriptors for cross-matching in re-ranking. We use the official implementation with the model trained on Pitts30k for the testing on the Pitts30k test set and the model trained on MSLS for others.

**CosPlace** [7][6]. This work casts the training of VPR as a classification problem and trains the VPR model on the San Francisco eXtra Large (SF-XL) datasets. We use the official VGG16 model (with the 512-dim output feature) for testing.

**MixVPR** [2][7]. This work introduces an innovative holistic feature aggregation approach, which utilizes feature maps extracted from the pre-trained backbone as global features. A stack of Feature-Mixer is employed to integrate global relationships into each feature map to yield global descriptors of place images. MixVPR is trained on the GSV-Cities [1] dataset and we conduct comparisons using the best configuration (ResNet50 with 4096-dim output features).

---

[2]https://github.com/mapillary/mapillary_sls
[3]https://github.com/Nanne/pytorch-NetVlad
[4]https://github.com/yxgeee/OpenIBL
[5]https://github.com/RuotongWANG/TransVPR-model-implementation
[6]https://github.com/gmberton/CosPlace
[7]https://github.com/amaralibey/MixVPR

**EigenPlaces** [10][8]. As an improvement of CosPlace, this work enhances the viewpoint robustness of learned global representations by training networks on images captured from various perspectives of the same places. EigenPlaces uses the CNN model as the backbone and attains superior performance on most VPR benchmark datasets. We adopt the official implementation and the best configuration based on the ResNet50 backbone to generate 2048-dim global descriptors for comparison.

**SelaVPR** [41][9]. The work introduces a novel hybrid global-local adaptation method for seamlessly adapting the pre-trained foundation model to the VPR task. This method only fine-tunes lightweight adapters without altering the pre-trained backbone, and the fine-tuned model can yield both global and local features for two-stage VPR. Additionally, a mutual nearest neighbor local feature loss is designed to train the local adaptation module. The produced dense local features are used in cross-matching for re-ranking, without time-consuming spatial verification. The official implementation with the model trained on Pitts30k is used for the testing on the Pitts30k test set, and the model trained on MSLS is used for others.

**CricaVPR** [39][10]. This work presents a cross-image correlation-aware representation learning method for VPR, which employs the attention mechanism to establish the correlation among multiple images within a batch. This method can leverage the cross-image variations as a cue to guide representation learning, and when producing the feature of each image, it can obtain useful information from other image features to improve its robustness against viewpoint variations, condition variations, and perceptual aliasing. To further enhance robustness, a multi-scale convolution-enhanced adaptation method is proposed in CricaVPR, adapting the pre-trained foundation model DINOv2 to the VPR task. We used the official implementation and configuration for comparison.

**SALAD** [27][11]. This method utilizes the pre-trained foundation model DINOv2 as the backbone, and treats the soft assignment in NetVLAD as an optimal transport problem, solving it using the Sinkhorn algorithm. This soft assignment takes into account both the relationships between features and clusters, as well as between clusters and features. Additionally, it introduces a "dustbin" cluster, which is aimed at selectively discarding features identified as non-informative, thereby improving the overall quality of descriptors. We used the official pre-trained DINOv2-SALAD model for the evaluation.

---

[8]https://github.com/gmberton/EigenPlaces

[9]https://github.com/Lu-Feng/SelaVPR

[10]https://github.com/Lu-Feng/CricaVPR

[11]https://github.com/serizba/salad

