# OpenReview forum: "SuperVLAD: Compact and Robust Image Descriptors for Visual Place Recognition"
_NeurIPS.cc/2024/Conference — NeurIPS 2024 poster_

### Official Review · Reviewer_w8L8 · 2024-07-01

**Soundness:** 3
**Presentation:** 3
**Contribution:** 2
**Rating:** 6
**Confidence:** 4

**Summary:**

The paper addresses the problem of learning a global image descriptor suitable for visual place recognition.
The work builds on top of NetVLAD, i.e. performs a soft assignment of local descriptors to learnable clusters, with the main difference that the model directly aggregates the local descriptors rather than residuals per cluster. This independence of the global embedding from cluster centers is argued to increase the robustness to domain shifts.

The retrieval performance is increased by the incorporation of related works, namely ghost clusters [61] that allow to ignore ambiguous local descriptors, and a cross-image transformer within a training batch [39] to obtain viewpoint stability.
With DINOv2 as backend features, the approach shows SotA performance, comparable to recent works from CVPR'24.

Experimental results demonstrate that only 4 clusters are sufficient, leading to a compact descriptor. The authors also propose a 1-cluster VLAD (which can be seen as a binary classification of local descriptors useful/non-useful) resulting on a 768 dim. descriptor and SotA performance compared to using GeM pooling [47] or the ViT/DINOv2 class token (both of dim. 768)

**Strengths:**

**S1** The approach demonstrates SotA retrieval performance at smaller global descriptor size (Table 2), while maintaining a comparable inference time.

**S2** The experimental evaluation of the approach is extensive and provides valuable comparisons to other SotA methods. Ablation studies document the influence of chosen architecture designs and the influence of different feature backends and training datasets well.
The proposed approach could have been presented more "powerful" be hiding some of those insights, e.g. Tab. 3 and Tab. 4. I appreciate this.

Note: The good performance at small descriptor size, and thus its wider applicability, is the main reason why I lean towards acceptance. Otherwise the technical contribution and novelty is small.

**Weaknesses:**

**W1**: The proposed architecture is motivated by the missing capability of NetVLAD to generalize well across domains (line 51ff). Table 3 tries to make this point, and shows that SuperVLAD performs better than NetVLAD on MSLS when trained on Pitts30k. Though, the reported improvement of SuperVLAD over NetVLAD of 5pp is negligible when DINOv2 is used as backbone (0-2pp). When NetVLAD is trained on a more versatile dataset (GSV-Cities) its performance improves by ~30pp and the gap between SuperVLAD and NetVLAD becomes negligible. This demonstrates that usage of a large scale, and versatile training data set is much more important than the modified local descriptor aggregation (one of the main contributions of the paper.)

[The remaining contribution is that SuperVLAD results in a more compact descriptor.]

**W2** The novelty of the proposed work is limited. The main notable difference to previous work is the direct aggregation of descriptors rather than residuals per cluster. Otherwise the approach is very close to NetVLAD and borrows the ideas from [61] for "ghost clusters" and  [39] for cross-image interaction to obtain more robust features. To achieve SotA performance ghost clusters are not relevant (Tab. 4) but the cross-image encoder is required to achieve the performance of [39] (which is fair) and [40]

**W3** The cherry picked results in Figure 4 do not provide much value, because such a matrix could be constructed in favor of any of the approaches. An illustration of typical failure cases of the proposed approach would be more helpful for the reader to understand the limitations of the approach.

**Questions:**

1) In the comparison in Table 5, what do you refer to as "class token"? If it's the ViT class token, is a fine-tuned on the task?

2) Figure 5 misses a point for CircaVPR, which has a Recall@1 of 94.9. What is its inference time?

3) To be able to compare all approaches 1:1 it would have been useful to use the same feature dimensionality for all of them, e.g. via training a final FC layer. Have experiments in this direction been conducted?

**Limitations:**

Limitations addressed adequately in the appendix. This discussion should be moved to the main paper.

---

> ### Author Rebuttal · Authors · 2024-08-06
>
> Thanks for your positive recommendation and constructive comments, as well as the recognition of our compact descriptor (especially for 1-cluster VLAD). We will move the limitations discussion to the main paper as you suggested. The following are Responses to the Questions (denoted as **Q**) and Weaknesses (denoted as **W**).
>
> **Q1. About “class token”:**
>
> The “class token” is output by the DINOv2 model, which can also be called ViT class tokens (DINOv2 is a ViT model trained on the large-scale curated LVD-142M dataset). It is also finetuned on GSV-Cities (VPR dataset). That is, all methods in Table 5 have the same settings (backbone, training setting, etc.) except for the descriptor, to compare the performance of descriptors fairly.
>
> **Q2. Inference time of CricaVPR:**
>
> The inference time of CricaVPR is 17.2ms (without PCA), which is slower than our SuperVLAD (13.5ms). We will add it to Figure 5. Besides, CricaVPR uses the optional PCA operation for dimensionality reduction, which requires more inference time if PCA is added.
>
> **Q3. Using the same feature dimensionality to compare via a final FC layer:**
>
> We agree with you that adding the final FC layer allows different methods to output features of the same dimensionality. However, the recent DINOv2-based works (e.g., SALAD and CricaVPR) do not do this. If we directly add a final FC layer to retrain the models, it will involve the issue of optimal training parameter setting, which may make the result much worse than the original works. In this case, it is also unfair to the other works. By the way, our method aims at producing compact descriptors without additional dimensionality reduction/ transformation techniques. So, we do not add a final FC layer, which ensures that our aggregation module is lightweight.
>
> **W1: Large-scale training dataset vs. the modified aggregation method**
>
> We agree with you that the large-scale training dataset can bring more improvements than the modified aggregation method for VPR in most cases. However, a robust aggregation algorithm is important when large training datasets are lacking. It is helpful to improve the performance floor in bad cases.
>
> **W2. The novelty of the proposed method:**
>
> It's important to note that NetVLAD is a very successful method but its domain generalization performance is still hampered by cluster centers. Our improvement is reasonably motivated and can address this import issue (also gets good results). Unlike previous improvements that usually complicate NetVLAD, our improvements simplify it by directly aggregating local features assigned to each cluster, which no longer computes residuals and is therefore free from the cluster centers (the insight is novel). That is, SuperVLAD is a simple yet effective method (to enhance the generalizability across different domains).
>
> **W3. The illustration of typical failure cases:**
>
> Thanks again for your suggestion. We provide detailed failure cases in the attached PDF (Figure 2), including cases where the retrieved image is geographically close to the query image but outside the 25-meter threshold, as well as some cases of challenging natural scenes without landmarks. We will add these failure cases to our final paper.

---

### Official Review · Reviewer_7MN7 · 2024-07-05

**Soundness:** 4
**Presentation:** 4
**Contribution:** 3
**Rating:** 6
**Confidence:** 5

**Summary:**

The authors proposed a new method called SuperVLAD for visual place recognition. The authors aims to fix the shortage of the previously mature NetVLAD, which is NetVLAD is not robust against domain gap and have to use high dimensional features.
SuperVLAD reduces the number of "clusters" used to smaller value and also propose a low-dimensional 1-Cluster VLAD descriptor, comparable in dimension to GeM pooling but with better performance. Experimental results show that SuperVLAD, when combined with a transformer-based backbone, outperforms NetVLAD, achieving better domain generalization and surpassing state-of-the-art methods on benchmark datasets.

**Strengths:**

The work made a small modification on the formulation of NetVLAD. The intuition is based on demonstrations in figure 1.
The number of learnable parameters is reduced by half, because finding the centroid ck is no longer a requirement.
The work is compatible with GhostVLAD which uses so-called ghost clusters to absorb useless scene elements.
The work compares with several new methods including ones from CVPR24. Its feature dimensionality used is the relatively lower ones while having the highest recall, which verifies one of the claims.
Resource consumption is reasonable.
Limitation and ablation analysis is comprehensive.

**Weaknesses:**

1. The novelty of the method is limited. It is a modification of the mature NetVLAD method and the modification appears to be small.
2. Performance improvement is less obvious considering table 2. SuperVLAD shows minor differences between the second best method.
3. To support the generalization claim, the authors need to show more comprehensive comparisons. How will the method perform if being used in dramatically different environments (such as training indoors but testing outdoors)? Indoor environments could be ScanNet or ETH3D, etc. So far, there isn’t an obvious domain gap between the datasets used in the generalization tests, and the generalization ability claim could be better justified.

**Questions:**

There are inconsistencies in dataset usage between experiments in 4.3 (table 2) and generalization tests (table 3). Why do the two tables report accuracy from different set of datasets?

**Limitations:**

The performance improvement mostly comes from the DINOv2 backbone, as explained in section 4.3. Table 3 shows that if NetVLAD uses the ViT backbone, the performance gap between NetVLAD and SuperVLAD is less obvious. The authors also confirmed such shortcomes in Appendix A. The SuperVLAD no longer outperformed the NetVLAD in commonly used settings.

Remaining suggestions not affecting rating:
It would be better to note the backbone features (e.g. DINOv2) in table 2.
When explaining the difference in scene coverage between datasets in the ablation study, it would be useful to include examples of the dataset scenes so that readers can visualize the data domain.

---

> ### Author Rebuttal · Authors · 2024-08-06
>
> Thanks for your positive recommendation and insightful comments. The following are Responses to the Question (denoted as **Q**) and Weaknesses (denoted as **W**).
>
> **Q. Inconsistencies in datasets between Table 2 and Table 3:**
>
> Sorry to confuse you. We compare our SuperVLAD with other methods on four benchmark datasets, as shown in Table 2. However, considering that we have a large number of ablation experiments, all ablation experiments, i.e., Table 3,4,5,6,7, are only conducted on the two most commonly used datasets (Pitts30k and MSLS) to show the performance more clearly. Besides, Pitts30k and MSLS have their own training and test/val sets, so we can simply use cross-validation (training on Pitts30k and testing on MSLS, and vice versa) to demonstrate the ability of SuperVLAD to address domain shift.
>
> **W1. The modification to NetVLAD:**
>
> We agree with you that NetVLAD is a mature method. However, the robustness against domain shift is still an important issue to be addressed. Our SuperVLAD is reasonably motivated and gets good results. Different from previous improved methods that usually complicate NetVLAD, our improvements simplify it to enhance the generalizability across different domains (the insight is novel). That is, our SuperVLAD is a simple yet effective method.
>
> **W2. The performance improvement:**
>
> Although the recognition performance improvement of our SuperVLAD over the previous SOTA method is not very obvious, our method is more lightweight than SOTA methods like SALAD. The number of parameters in the SuperVLAD aggregator (0.0038M) is less than 3/1000 that of the SALAD aggregator (1.4M). The output descriptor dimension of SuperVLAD (3072) is less than 1/2 that of SALAD (8448).
>
> **W3. The experiments on the indoor dataset:**
>
> Thanks for your suggestion. We use the trained models (on GSV-Cities) to test on the indoor dataset Baidu Mall [1][2] to support the generalization claim. SuperVLAD achieves better results than NetVLAD (DINOv2-based), as well as other SOTA methods on this dataset (see the following two tables), which further supports our claim.
>
> | Method | R@1 / R@5 / R@10|
> |  :----:  | :----: |
> |NetVLAD |68.5 / 81.2 / 86.5|
> |SuperVLAD |69.5 / 82.7 / 87.3|
>
> | Method | R@1 / R@5 / R@10|
> |  :----:  | :----: |
> |CricaVPR |61.3 / 78.5 / 85.9|
> |SelaVPR |66.6 / 79.6 / 85.9|
> |SALAD |68.3 / 82.1 / 86.8|
> |SuperVLAD |69.5 / 82.7 / 87.3|
>
> **For the suggestion in Limitations:**
>
> Thanks again for your suggestion. We provide examples of datasets to show the data domain gap between Pitts30k and MSLS in the attached PDF (Figure 1). We will add it to our final paper.
>
> **References**
>
> [1] Sun, Xun, et al. "A dataset for benchmarking image-based localization." Proceedings of the IEEE Conference on Computer Vision and Pattern Recognition. 2017.
>
> [2] Keetha, Nikhil, et al. "Anyloc: Towards universal visual place recognition." IEEE Robotics and Automation Letters (2023).

---

> > ### Comment · Reviewer_utGd · 2024-08-11
> >
> > Thanks Reviewer 7MN7 for asking about domain gap on indoor environments, and authors for running the experiment. While not familiar with BaiduMall of Sun et al (thanks for mentioning it!) - it seems suitable for the task of localization, while ScanNet was not designed for localization tasks (but could probably be split into query and db images).
> > Potential candidates could also be (maybe for future work)
> > * InLoc: Indoor Visual Localization with Dense Matching and View Synthesis, of Taira et al
> > * Matterport3D: Learning from RGB-D Data in Indoor Environments of Chang et al  (might need a split for query images).

---

### Official Review · Reviewer_wUjJ · 2024-07-06

**Soundness:** 3
**Presentation:** 3
**Contribution:** 2
**Rating:** 6
**Confidence:** 5

**Summary:**

In this paper, the authors focus on reducing the dimension of NetVLAD method for the task of visual place recognition. More specifically, the proposed method named SuperVLAD combines previous techniques including the powerful DIVOv2 as backbone, free from clusters, and ghost clusters. Experiments demonstrate that the proposed method outperforms pervious methods on Pitts30k, MSLS-val, Nordl, and SPED datasets. Moreover, the extensive ablation studies are conducted the verify the effectiveness of different components.

**Strengths:**

1.	Good performance. Although the proposed method is mainly based on previous techniques, such as a more powerful backbone, ghost clusters, cross-image interaction, et al, the proposed method outperforms almost all previous methods on four public datasets. This sets a new benchmark and is helpful in real applications.

2.	Extensive ablation studies. In addition to the comparisons to previous approaches, the authors conduct extensive ablation studies on the backbone, training set, ghost cluster, pooling strategy, and the number of the clusters. These experiments help the readers understand which techniques contribute most to the improvements against other methods.

3.	The paper is well written and easy to read.

**Weaknesses:**

My concern is about the comparison between the proposed SuperVLAD and NetVLAD.

1.	According to the Table 7 in the appendix. Table 7 shows that with DINOv2 as backbone, when increasing the number of clusters from 4 to 64, NetVLAD does not improve the performance. Conversely, it loses the accuracy. This is not consistent with the claim of the paper that the proposed approach reduces the dimension of features given by NetVLAD as even with very smaller number of clusters, NetVLAD also has very good performance as SuperVLAD does. I am very curious what will happen if the number of clusters for NetVLAD and SuperVLAD is set to 1, 2, 3.

2.	According to Table 4 and 6, the ghost cluster does not improve the performance when a the powerful DINOv2 is used as the backbone. Cross-image encoder contributes most to the improvements. It would be great to show the results of NetVLAD with 4 clusters, DINOv2, and cross-image encoder. Then we will know how many improvements come from other techniques used in SuperVLAD.

**Questions:**

NA

**Limitations:**

Yes

---

> ### Author Rebuttal · Authors · 2024-08-06
>
> Thanks for your valuable comments and suggestions. We hope the following clarifications will be able to address your concerns.
>
> **W1. About the number of clusters:**
>
> Sorry to confuse you.
>
> -	First, since we did not specifically emphasize that as the capabilities of the backbone model improve (e.g., DINOv2) and the amount and quality of training data increase (e.g., GSV-Cities), both NetVLAD and SuperVLAD can achieve excellent performance even with a small number of clusters, this may lead readers to misunderstand that it is unique to SuperVLAD. We will improve it in the revised paper. However, when using simple models or smaller training datasets, the performance of a small number of clusters is not as good as that of a large number of clusters, and our SuperVLAD can outperform NetVLAD with a small number of clusters, as shown in Table 7. Besides, to our best knowledge, our work is the first attempt to set the number of clusters to be very small (only 4).
>
> -	Second, for setting the number of clusters to 1, 2, and 3, we provide the following explanations and experimental results. Setting the number of clusters to 1 will make the softmax layer for soft assignment meaningless. Our 1-cluster VLAD is achieved by adding additional ghost clusters, and there is not just one cluster during soft assignment. In fact, if there is only one cluster, all features are assigned to one category and the weight is "1". So SuperVLAD directly sums up all local features, which is equivalent to global average pooling (multiply by a constant), while NetVLAD subtracts a constant vector from the SuperVLAD vector (equivalent to translating the feature space), which does not affect the similarity calculation, i.e., the retrieval results are consistent. As for setting 2 or 3 clusters, the conclusion is the same as that in Table 7. That is, for using the powerful backbone (DINOv2) and the large-scale GSV-Cities dataset, the recognition performance of SuperVLAD and NetVLAD is nearly equal (but our SuperVLAD is simpler and more lightweight). However, for training on smaller datasets like Pitts30k, or/and using the simpler backbone like CCT, our SuperVLAD can outperform NetVLAD on recognition performance with an obvious margin. The detailed results (R@1/R@5/R@10) are shown in the following tables. Both NetVLAD and SuperVLAD are without ghost clusters and the cross-image encoder for fair comparison.
>
> ①	 When training DINOv2-based models on GSV-Cities, both NetVLAD and SuperVLAD achieve good results without obvious differences on the R@N metric.
> |2 clusters   | Pitts30k | | | MSLS-val |
> |:---|:---:|:---:|:---:|:---:|
> |NetVLAD    | 91.8 / 96.1 / 97.3 | | | 91.5 / 96.5 / 96.9|
> |SuperVLAD  | 92.4 / 96.6 / 97.5 | | |  91.2 / 96.2 / 96.8|
>
> |3 clusters   | Pitts30k | | | MSLS-val |
> |:---|:---:|:---:|:---:|:---:|
> |NetVLAD    | 91.4 / 96.2 / 97.5  | | |  91.8 / 95.8 / 96.8|
> |SuperVLAD  | 92.1 / 96.7 / 97.8  | | |  91.9 / 95.9 / 96.2|
>
>    ② When training CCT-based models on Pitts30k, SuperVLAD outperforms NetVLAD, with an obvious margin on Pitts30k and an even larger margin on MSLS (with 12.3% absolute R@1 improvement on MSLS-val with 3 clusters). This shows that our SuperVLAD performs better than NetVLAD with a simple backbone and small training dataset, i.e., the lower limit of performance is higher than NetVLAD. Besides, it once again verifies that SuperVLAD is more robust than NetVLAD when there is a domain shift in training data and test data (i.e., training on Pitts30k and testing on MSLS). By the way, 2 and 3 clusters get obviously worse results than 8 clusters (see Table 3 in paper).
>
> |2 clusters   | Pitts30k | | | MSLS-val |
> |:---|:---:|:---:|:---:|:---:|
> |NetVLAD    | 78.3 / 90.5 / 93.5  | | |  42.4 / 53.4 / 57.2|
> |SuperVLAD  | 80.1 / 90.9 / 93.9  | | | 48.0 / 61.9 / 65.5|
>
> |3 clusters   | Pitts30k | | | MSLS-val |
> |:---|:---:|:---:|:---:|:---:|
> |NetVLAD   |  79.5 / 90.7 / 93.9  | | |  43.1 / 55.9 / 60.7|
> |SuperVLAD  |  81.8 / 91.4 / 93.8 | | |  55.4 / 67.8 / 72.2|
>
> From the above results, we can summarize the advantages of SuperVLAD as follows:
>
> a. When using a powerful backbone and a large-scale training dataset, SuperVLAD achieves the nearly same performance as NetVLAD using fewer (about half) parameters.
>
> b. When using a simple backbone and a small training dataset, SuperVLAD can outperform NetVLAD by an obvious margin.
>
> c. When there is a domain gap in training data and test data, SuperVLAD is more robust than NetVLAD.
>
> **W2. Results of NetVLAD with 4 clusters, DINOv2, and cross-image encoder:**
>
> -	First, we agree with you that the ghost cluster does not improve the performance when we use the powerful DINOv2 backbone. Considering that the ghost cluster can bring improvement (by eliminating useless information) when using a simple backbone (as shown in Table 4 of our paper), we adopted it in SuperVLAD.
> -	Second, we also agree with you that showing the results of NetVLAD with 4 clusters, DINOv2, and cross-image encoder can better analyze the improvements from other techniques used in SuperVLAD. The results (trained on GSV-Cities) are shown in the following table. After adding other techniques to both SuperVLAD and NetVLAD (denoted as NetVLAD*), SuperVLAD can outperform NetVLAD on recognition performance, i.e., SuperVLAD is more suitable for adding these technologies. (We have shown in our paper that the recognition performance of DINOv2-based SuperVLAD and NetVLAD is comparable without adding other technologies, in which case the advantage of SuperVLAD is simpler, more lightweight, and robust to domain gap.)
>
> |Method   | Pitts30k | | MSLS-val | | Nordland | | SPED|
> |:---|:---:|:---:|:---:|:---:|:---:|:---:|:---:|
> |NetVLAD*  |93.8 / 97.0 / 98.0 | | 91.1 / 96.8 / 97.3 | | 89.6 / 95.9 / 97.4 | | 92.6 / 96.7 / 97.5|
> |SuperVLAD  |95.0 / 97.4 / 98.2 | | 92.2 / 96.6 / 97.4 | | 91.0 / 96.4 / 97.7 | | 93.2 / 97.0 / 98.0|
>
> Thanks again for your thoughtful review, and please let us know if you have further questions.

---

> > ### Author Response · Authors · 2024-08-12
> > **Official Comment by Authors**
> >
> > Dear Reviewer wUjJ,
> >
> > Thanks again for your helpful comments and suggestions. We hope our responses could address your concerns. As the discussion period nears its end, please let us know if you have further questions or concerns. We'd be very glad to address them.
> >
> > Best regards,
> >
> > Authors

---

> > ### Comment · Reviewer_wUjJ · 2024-08-14
> > **post rebuttal**
> >
> > Thanks for the rebuttal. I have no other concerns and upgrade my rating.

---

> > > ### Author Response · Authors · 2024-08-14
> > > **Thanks for your reply**
> > >
> > > We are pleased that our response has addressed your concerns. Many thanks for increasing the score.

---

### Official Review · Reviewer_utGd · 2024-07-10

**Soundness:** 3
**Presentation:** 3
**Contribution:** 3
**Rating:** 6
**Confidence:** 3

**Summary:**

The paper introduces SuperVLAD for addressing Visual Place Recognition (VPR) task. The proposed SuperVLAD descriptor is an improvement to NetVLAD descriptor, by reducing the dimensionality, by using fewer clusters and

**Strengths:**

* Simple but effective method, reducing the number of parameters in VLAD (by reducing the number of clusters and removing the cluster centers influence).
* In dept comparison with other (especially recent methods)
* Achieving results on-par with NetVLAD (Tab 3) with fewer parameters.

**Weaknesses:**

1. It is a bit difficult to understand what contributes to the results; there is a table (Tab 6) comparing the influence of number of clusters, and comparison on backbones (e.g. Tab 3). It would be nice to have a number of clusters column in Tab3 - to show the influence of number of clusters for a given backbone, i.e. help understand where the improvements come from.
From Tab 3 -- the results, on the same backbone, seem on par with NetVLAD -- it might be worth emphasizing that this is achieved with fewer parameters, due to the smaller number of cluster centers.
2. What is the impact of discarding cluster centers? It is not clear from the text / tables if discarding the cluster centers influences the results or not.
3. Evaluation on SPED dataset (lower quality images) might benefit even more from adding ghost cluster (Tab 4).
4. Missing evaluation on Pittsburgh-250k and Tokyo 24/7 -- How does SuperVLAD perform for changing environment conditions, such as day-night?
5. MixVPR achieves comparable results on Pitts30k -- with a much smaller architecture (ResNet-50). In Tab2 -- no mention on what data are the models trained.

**Questions:**

1. Results in Table 2 (NetVLAD) don't match the results in SALAD paper ([27] -- also wrongly cited as [40] in Tab 2). Which are the correct ones?
2. How would SuperVLAD perform if the backbone was ResNet-50? ( to make it easier to compare with existing methods, like MixVPR[2])

(minor) remark: adding Venue column in Tab 2 doesn't add any value to the evaluation -- might be worth using the space to add the backbone for the specific method, to make comparison easier.

3. What are the total number of parameters and number of trainable parameters of the methods compared in Tab2?

**Limitations:**

Somewhat unclear ablation -- information is present, but not aggregated, making it difficult to understand the contribution of individual components (e.g. removing cluster centers, reducing the number of clusters -- this could be a nice plot / individual rows in a table), adding ghost clusters, changing backbone -- i.e. highlighting the contribution of each change individually.

---

> ### Author Rebuttal · Authors · 2024-08-06
>
> Thanks for your insightful comments and valuable suggestions. The following are Responses to the Questions (denoted as **Q**) and Weaknesses (denoted as **W**).
>
> **Q1. Results of NetVLAD in Table 2:**
>
> Both of our results and the results in the SALAD paper are correct. We use the original version of NetVLAD as described in Appendix G (also can be seen in NetVLAD paper), i.e., VGG16 backbone and trained on Pitts30k. The results of DINOv2-based NetVLAD trained on GSV-Cities (basically the same as in SALAD) are shown in Table 3 and Table 7 of our paper, which basically matches the results in SALAD. Thank you for pointing out our wrong citation in Table 2.
>
> **Q2. SuperVLAD perform with ResNet-50:**
>
> Our SuperVLAD with ResNet-50 cannot outperform MixVPR. To take full advantage of SuperVLAD we need to use the Transformer-based backbone, which is a limitation of SuperVLAD and has been discussed in Appendix A (Limitations) and Appendix C (Results with VGG16). Considering that more and more VPR methods are using Transformer models and even the DINOv2 model, we can also use the DINOv2 backbone uniformly for fair comparison. There is a DINOv2-based MixVPR proposed, i.e. DINO-Mix [1], and SuperVLAD outperforms DINO-Mix with an obvious margin.
>
> | Method |  Pitts250k | Pitts30k | Tokyo24/7|
> |:---|:---:|:---:|:---:|
> |DINO-Mix   | 94.6 | 92.0 | 91.8|
> |SuperVLAD  | 97.2 | 95.0 | 95.2|
>
> **Q3. Total number of parameters and trainable parameters:**
>
> We appreciate your attention to the details in Table 2. It's important to note that the methods compared in this table use different backbones, which may impact the direct comparability of the number of parameters. To address this concern, we provide a comparison of SALAD and SuperVLAD that both use the DINOv2-base backbone and only finetune the last four Transformer blocks of the backbone. The value in parentheses is the number of parameters in the optional across-image encoder.
>
> | Method |  Total (M)  |  Trainable (M)  |  aggregator (M)|
> |:---|:---:|:---:|:---:|
> |SALAD     |   88.0        |  29.8       |   1.4 |
> SuperVLAD  |  86.6 (+11.0)  |  28.4 (+11.0) |  0.0038|
>
> The number of parameters in the aggregator is the most important consideration. The number of parameters in our aggregator is much lower than that of SALAD (less than 3/1000). As far as we know, the SuperVLAD aggregator has a smaller number of parameters than any other common methods except global pooling methods (e.g., GeM).
>
> **W1. About the readability of the results:**
>
> Thanks for your suggestion to improve the readability of our paper. We will improve it in the final paper.
> We have introduced the number of clusters in the table caption of Table 3 and we will move it to the table. For the same backbone, the improvement of SuperVLAD mainly exists when there is a gap between the training data and the test data (e.g., training on Pitts30k and testing on MSLS, see the reply to **W2** for more details), which is also fits our motivation. In addition, we use fewer parameters than NetVLAD not only because of the smaller number of clusters, but also by removing the cluster centers, which reduces the parameters of NetVLAD by about half even with the same number of clusters. We will emphasize this point as you suggested.
>
> **W2. The impact of discarding cluster centers:**
>
> Discarding cluster centers can improve robustness when there is a domain gap between the data in training and inference. As shown in Table 3, when the models trained on Pitts30k (only urban images) are tested on MSLS (including urban, suburban, and natural images), our SuperVLAD outperforms NetVLAD. More specifically, discarding cluster centers brings an absolute R@1 improvement of 6.8% (using CCT backbone) and 2.4% (using DINOv2) on MSLS-val. However, for the models trained on the GSV-Cities dataset, there is little difference in results between SuperVLAD and NetVLAD because GSV-Cities covers diverse training data (i.e., no obvious gap between training and test data). Please refer to Lines 312-324 in the paper for more details.
>
> **W3. Adding the evaluation on the SPED dataset in Tab 4:**
>
> The results on SPED with or without (denoted as “-ng” suffix) ghost cluster are shown in the following table. There is no obvious difference between them. Considering that it can eliminate useless information and bring improvements on some datasets, we still choose to keep it. More importantly, we use it to achieve the 1-cluster VLAD.
>
> | Method |  R@1  |  R@5  |  R@10 |
> |:---|:---:|:---:|:---:|
> |CCT-SuperVLAD-ng     | 64.7 |79.1 |84.2|
> |CCT-SuperVLAD        | 64.7 |78.6 |83.9|
> |DINOv2-SuperVLAD-ng  | 89.8 |94.7 |95.9|
> |DINOv2-SuperVLAD     | 90.1 |95.6 |95.9|
>
> **W4. Evaluation on Pittsburgh-250k and Tokyo 24/7:**
> The results on Pitts250k and Tokyo24/7 are shown in the following table. SuperVLAD also achieves SOTA results, showing good performance in day-night changes.
>
> |Method   | Tokyo24/7 | | | pitts250k |
> |:---|:---:|:---:|:---:|:---:|
> |SelaVPR    |  94.0 / 96.8 / 97.5    | | |  95.7 / 98.8 / 99.2|
> |SALAD     |  94.6 / 97.5 / 97.8  | | |  95.1 / 98.5 / 99.1|
> |SuperVLAD |  95.2 / 97.8 / 98.1   | | | 97.2 / 99.4 / 99.7|
>
> **W5. a. About MixVPR with ResNet50 and b. training data of models in Table 2:**
>
> -	a. Please see the response to **Q2**.
> -	b. The training data for MixVPR, CricaVPR, SALAD and our SuperVLAD is GSV-Cities (as described in Section 4.3), and for other methods is described in Appendix G.
>
> Thanks again for your thoughtful review, and please let us know if you have further questions.
>
> **References**
>
> [1] Huang, Gaoshuang, et al. "Dino-mix: Enhancing visual place recognition with foundational vision model and feature mixing." arXiv preprint arXiv:2311.00230 (2023).

---

> > ### Comment · Reviewer_utGd · 2024-08-11
> >
> > Many thanks to the authors for the detailed and thorough answers to all the questions and concerns raised by myself and the other reviewers.
> >
> > It would be worth including the answer to Q3 in the main paper, as it highlights a strength over SALAD ( the much smaller number of parameters in the aggregator ).
> >
> > After reading through the other reviews and the answers, I am editing my initial review to increase the rating to 6.

---

> > > ### Author Response · Authors · 2024-08-12
> > > **Thank you for your reply**
> > >
> > > Thanks a lot for your positive feedback and increasing the score! We'll be sure to incorporate your suggestions into our final paper.

---

### Official Review · Reviewer_TmXY · 2024-07-18

**Soundness:** 3
**Presentation:** 3
**Contribution:** 3
**Rating:** 7
**Confidence:** 4

**Summary:**

This work introduces SuperVLAD, a novel image descriptor for Visual Place Recognition. It eliminates the need for cluster centers, addressing NetVLAD's performance degradation due to data bias while being more lightweight. SuperVLAD generates compact descriptors with fewer clusters. Additionally, a 1-cluster VLAD method employs supernumerary ghost clusters for soft assignment, producing low-dimensional features that outperform same-dimensional GeM features or class tokens. Extensive experiments with transformer-based backbones demonstrate the effectiveness of this approach.

**Strengths:**

This work is well-written and easy to follow. The method and framework are technically robust and novel, offering a fresh perspective on image descriptor design for Visual Place Recognition. SuperVLAD achieves state-of-the-art performance across several place recognition benchmarks, demonstrating its effectiveness and competitiveness in the field. By eliminating the need for cluster centers, SuperVLAD addresses NetVLAD's performance degradation due to data bias while being more lightweight, which is crucial for real-time applications and resource-constrained environments. SuperVLAD generates compact descriptors with fewer clusters, leading to reduced memory and computational requirements. The introduction of a 1-cluster VLAD method employing ghost clusters for soft assignment is a novel approach, producing low-dimensional features that outperform same-dimensional GeM features or class tokens. Extensive experiments with transformer-based backbones highlight SuperVLAD's versatility and adaptability, leveraging state-of-the-art architectures to enhance visual place recognition performance. The method's broad applicability, including in autonomous navigation, robotics, and augmented reality, makes it a valuable contribution to the field. Additionally, the paper provides a comprehensive evaluation of SuperVLAD's performance, validating its effectiveness and establishing a solid foundation for its claims.

**Weaknesses:**

Provide more technical and experimental details. Please see my questions below The paper could benefit from more technical and experimental details. Please see my questions below.

**Questions:**

1.	How do these new image descriptors integrate with visual localization and SLAM systems? Specifically, it would be beneficial to understand how SuperVLAD can improve the accuracy and robustness of these systems in real-world applications.
2.	Please provide some efficiency results on SuperVLAD, such as inference time and training time. Additionally, information on the training data used would be helpful. Understanding the computational demands and data requirements is crucial for assessing the practical applicability of the method.
3.	To attract readers in a machine learning conference, it would be valuable to offer general insights into how deep learning can contribute to spatial scene perception. Discussing the importance of this task to the machine learning community and the broader implications of improving visual place recognition through advanced descriptors like SuperVLAD would enhance the paper's appeal.

**Limitations:**

please see my questions. Since this work is within a subarea in CV, please provide more insights into how ML can impact this area and how this application is important to ML.

---

> ### Author Rebuttal · Authors · 2024-08-06
>
> Thanks for your positive recommendation and encouraging words. The following are Responses to the Questions.
>
> **Q1. Integrate SuperVLAD with visual localization and SLAM systems:**
>
> Since SuperVLAD is a general visual place recognition (VPR) method, it can be used for visual localization and SLAM systems like previous standard VPR methods such as NetVLAD.
> For SLAM systems, our SuperVLAD can be used as the loop-closure detection method to rectify the accumulated mapping error. As you know, our SuperVLAD is more lightweight and can produce compact descriptors, so it is more suitable for resource-constrained environments like mobile robots SLAM. Given the good recognition performance of SuperVLAD, it may not require spatial consistency verification in loop-closure detection, which requires us to further explore its recall performance at 100% precision in future work. For visual localization, it usually only requires the VPR method to retrieve top-N candidate images (obtain location hypotheses), then perform local matching and compute 6-DoF pose [1]. Our experiments have shown that SuperVLAD can achieve good Recall@N performance and is therefore suitable for visual localization.
>
> **Q2. Experimental details: inference time, training time, and training data:**
>
> The experimental details including training data (i.e., GSV-Cities) and some other settings are described in Section 4.2 and Appendix D. The inference time is 13.5ms for a single image, as shown in Section 4.3 (Figure 5). The training time of our SuperVLAD is 81.6 minutes (training for 7 epochs on GSV-Cities) using two NVIDIA GeForce RTX 3090 GPUs, compared to 210 minutes for CricaVPR [2], which uses the same backbone and training dataset as ours. We will add the training time to the final paper.
>
> **Q3. How this application is important to ML, how ML can impact this area and what can SuperVLAD bring:**
>
> This is a very meaningful question. We should think more about the impact that the VPR task and our method (SuperVLAD) can bring to the ML community.
>
> -	First, the VPR research is a foundational part of studying how machine learning models perceive, understand, and represent geographic space.
> -	Second, the emergence of better VPR methods/models requires unifying the efforts of the machine learning, computer vision, and robotics communities. Especially when a novel machine learning model is proposed, it usually brings new solutions to the VPR task.
> -	Last, as a type of aggregation algorithm, VLAD-related methods can not only be used to aggregate image local features, but are also widely used for video sequences aggregation [3][4], speech clustering [5], and multi-modal information processing [6][7]. As a pure aggregation algorithm, our SuperVLAD does not add any prior information related to the VPR task. It also has the potential to be applied to other fields to aggregate other data. From this perspective, it is not just a vision algorithm but can be viewed as a machine learning algorithm. As for the broader implications of improving VPR through our SuperVLAD, we have discussed it in Appendix B.
>
> Thanks again for your meaningful questions.
>
> **References**
>
> [1] Sarlin, Paul-Edouard, et al. "From coarse to fine: Robust hierarchical localization at large scale." Proceedings of the IEEE/CVF conference on computer vision and pattern recognition. 2019.
>
> [2] Lu, Feng, et al. "CricaVPR: Cross-image Correlation-aware Representation Learning for Visual Place Recognition." Proceedings of the IEEE/CVF Conference on Computer Vision and Pattern Recognition. 2024.
>
> [3] Xu, Youjiang, et al. "Sequential video VLAD: Training the aggregation locally and temporally." IEEE Transactions on Image Processing 27.10 (2018): 4933-4944.
>
> [4] Naeem, Hajra Binte, et al. "T-VLAD: Temporal vector of locally aggregated descriptor for multiview human action recognition." Pattern Recognition Letters 148 (2021): 22-28.
>
> [5] Hoover, Ken, et al. "Putting a face to the voice: Fusing audio and visual signals across a video to determine speakers." arXiv preprint arXiv:1706.00079 (2017).
>
> [6] Wang, Xiaohan, Linchao Zhu, and Yi Yang. "T2vlad: global-local sequence alignment for text-video retrieval." Proceedings of the IEEE/CVF conference on computer vision and pattern recognition. 2021.
>
> [7] Wang, Hui, et al. "Pairwise VLAD interaction network for video question answering." Proceedings of the 29th ACM international conference on multimedia. 2021.

---

### Author Rebuttal · Authors · 2024-08-06

We sincerely thank all the reviewers for their valuable time and constructive comments on our work. We reply to the concerns of each reviewer individually and will incorporate the suggestions in the revised paper. We also attach a PDF containing figures, which we refer to when answering specific questions.

---

### Decision · Program_Chairs · 2024-09-25

**Decision:**

Accept (poster)

**Comment:**

This paper received unanimous acceptance from 5 reviewers (6,6,6,6,7) and, therefore a recommendation of acceptance.


Below are some concerns from AC, although they are **not** taken into account during the decision phase.

-------
The major contribution of this paper is a new formulation of the SuperVLAD layers, which are illustrated in Eq (4).
However, it seems that Eq (4) is just a simple case of Attention:
Given the well-known attention mechanism: $\rm{Attention}(q, k, v) = \rm{softmax}(q^T k) v$.
Equation (4) might be a special case of Attention by taking the query (q) as value (v) (q=v), that is: $\rm{softmax}(v^T k) v$, although there are some minor changes.
In this case, the major contribution, that is the formulation of the SuperVLAD layers, may not be valid anymore, thus the entire story (of removing cluster centers) in the introduction is not valid. The AC won't make a rejection recommendation based on this assumption, but the AC highly encourages the authors to (1) explain the differences between SuperVLAD and Attention in the final version or (2) withdraw the manuscript if the authors cannot provide strong justification for the difference or didn't notice this potential issue before.